# Dynamic modulation of subthalamic nucleus activity facilitates adaptive behavior

**Damian M. Herz** [1,2]*, **Manuel Bange**[2], **Gabriel Gonzalez-Escamilla**[2], **Miriam Auer**[2], **Muthuraman Muthuraman**[2,3], **Martin Glaser**[4], **Rafal Bogacz**[1], **Alek Pogosyan**[1], **Huiling Tan**[1], **Sergiu Groppa**[2‡], **Peter Brown**[1‡]

**1** MRC Brain Network Dynamics Unit at the University of Oxford, Nuffield Department of Clinical Neurosciences, University of Oxford, Oxford, United Kingdom, **2** Movement Disorders and Neurostimulation, Department of Neurology, Focus Program Translational Neuroscience (FTN), University Medical Center of the Johannes Gutenberg-University Mainz, Mainz, Germany, **3** Neural Engineering with Signal Analytics and Artificial Intelligence, Department of Neurology, University Hospital of Wuerzburg, Wuerzburg, Germany, **4** Department of Neurosurgery, University Medical Center of the Johannes Gutenberg-University Mainz, Germany

‡ These authors are joint senior authors on this work.
* damian.m.herz@gmail.com

**Data Availability Statement:** At present, participant consent does not allow for depositing

## Abstract

Adapting actions to changing goals and environments is central to intelligent behavior. There is evidence that the basal ganglia play a crucial role in reinforcing or adapting actions depending on their outcome. However, the corresponding electrophysiological correlates in the basal ganglia and the extent to which these causally contribute to action adaptation in humans is unclear. Here, we recorded electrophysiological activity and applied bursts of electrical stimulation to the subthalamic nucleus, a core area of the basal ganglia, in 16 patients with Parkinson's disease (PD) on medication using temporarily externalized deep brain stimulation (DBS) electrodes. Patients as well as 16 age- and gender-matched healthy participants attempted to produce forces as close as possible to a target force to collect a maximum number of points. The target force changed over trials without being explicitly shown on the screen so that participants had to infer target force based on the feedback they received after each movement. Patients and healthy participants were able to adapt their force according to the feedback they received ($P < 0.001$). At the neural level, decreases in subthalamic beta (13 to 30 Hz) activity reflected poorer outcomes and stronger action adaptation in 2 distinct time windows ($P_{cluster-corrected} < 0.05$). Stimulation of the subthalamic nucleus reduced beta activity and led to stronger action adaptation if applied within the time windows when subthalamic activity reflected action outcomes and adaptation ($P_{cluster-corrected} < 0.05$). The more the stimulation volume was connected to motor cortex, the stronger was this behavioral effect ($P_{corrected} = 0.037$). These results suggest that dynamic modulation of the subthalamic nucleus and interconnected cortical areas facilitates adaptive behavior.

the full original dataset. A minimum example dataset and all code is available on https://data.mrc.ox.ac.uk/data-set/subthalamic-nucleus-correlates-force-adaptation (doi: 10.5287/ora-9ovjdypbb). Source data are provided with this paper as supplemental data.

**Funding:** DMH is supported by a postdoctoral grant from the Independent Research Fund Denmark (0168-00014B). AP, HT and PB are supported by the Medical Research Council (MC_UU_00003/2). RB is supported by the Medical Research Council (MC_UU_00003/1). This research was funded by the UKRI [MC_UU_00003/2]. The funders had no role in study design, data collection and analysis, decision to publish, or preparation of the manuscript.

**Competing interests:** The authors have declared that no competing interests exist.

**Abbreviations:** AUC, area under the curve; BIC, Bayesian information criterion; CT, computerized topography; CV, coefficient of variation; DBS, deep brain stimulation; HC, healthy control; LFP, local field potential; LME, linear mixed-effect; MVC, maximum voluntary contraction; PD, Parkinson's disease; RMSE, root mean squared error; STN, subthalamic nucleus.

## Introduction

To successfully navigate between affordances offered by our environment, we have to learn how actions differ regarding their usefulness and update these associations if they no longer lead to desirable outcomes [1–3]. This does not only apply to deciding what action to choose, but also how to perform it. For example, during foraging agents do not only need to choose between, e.g., eating grapes or nuts depending on their nutritional value, but also learn how much force to apply for cracking a nut without crushing it. In the reinforcement-learning framework, action-value associations are learned by comparing actual and expected value for different options and then adapting actions accordingly [3–5].

Neurobiologically, there is strong evidence that firing rates of dopaminergic midbrain neurons and consequent striatal dopamine release are modulated depending on the difference between the actual and expected value [6–9]. With the coincident presence of glutamate in striatal synapses, this signal is thought to alter excitability of the respective cortical neurons and induce plasticity mechanisms through the cortico-basal ganglia pathways [3,10–14]. Thus, neural activity patterns leading to successful outcomes will be strengthened making the action more likely in the future, while unsuccessful actions should be modified. The electrophysiological correlates of these processes in the basal ganglia and the extent to which they causally contribute to action adaptation in humans is not well understood.

Patients with Parkinson's disease (PD) undergoing deep brain stimulation (DBS) offer the unique opportunity to directly record electrophysiological activity from the subthalamic nucleus (STN) and apply invasive electrical stimulation in humans. The STN is a core part of the indirect (and hyperdirect) basal ganglia pathway exerting a net inhibitory effect on thalamo-cortical connections. Previous STN recordings in PD patients have shown that movement-related activity in the beta-band (approximately 13 to 30 Hz) is strongly modulated during force production [15–18]. Importantly, beta activity is reduced when force is adjusted irrespective of whether this entails an increase or a decrease in force [19]. In other words, decreases in beta activity appear to be reflective of changes in force rather than force per se. A relationship between STN beta activity and concomitant movement changes has also been observed after the movement is terminated. For example, it has been demonstrated that movement errors due to external perturbations reduce STN beta activity compared to correct movements, but only if this error is relevant for action adaptation [20]. Another study found that STN beta power after the movement is relatively higher when an action outcome has been favorable and the action should be repeated and relatively lower after suboptimal actions necessitating change [21].

However, in these previous studies, it was already obvious to the participants how the movement should be adjusted at the time when they observed a difference between actual and expected outcome conflating action evaluation and preparation. In other words, the observed changes might primarily have reflected movement preparation of the next movement rather than evaluation of the previous movement. Furthermore, it remains elusive whether these STN activity changes are merely correlative in nature or if they causally contribute to action evaluation and adaptation. This is particularly important since current approaches of therapeutic adaptive DBS employ STN beta activity as a feedback signal, which in turn is modulated by stimulation [22,23].

To address these outstanding issues, we conducted electrophysiological recordings and applied bursts of electrical STN stimulation in 16 PD patients who had undergone DBS surgery and were treated with dopaminergic medication. Based on a previous study of decision-making under uncertainty [24], we designed a force adaptation task in which participants continuously had to adapt their grip force based on the value associated with their previous action

reflecting how close the actual force was to the target force. Importantly, the Value-feedback was dissociated from a second feedback cue necessary for action adaptation, so that patients could not infer with certainty how to adapt their force until after the second cue. We hypothesized that STN beta activity would be modulated by action-value feedback and consequent adaptation of grip force. Moreover, we expected extrinsic modulation of beta activity through direct electrical stimulation to modify trial-to-trial action adaptation.

## Results

During the task, participants attempted to produce forces as close as possible to a target force in order to collect a maximum number of points. While they were aware of the approximate target force level on the first trial (approximately 20% to 25% of their maximum voluntary contraction, MVC), the target force changed over trials without being explicitly shown on the screen so that participants had to infer target force based on the feedback they received after each movement. The first feedback cue (Value-feedback) indicated how close the actual force was to the target force (ranging from 0 (worst) to 10 (best)) and the second feedback cue (Direction-feedback) showed whether the force had been too low or too high (Fig 1A). Thus, after the Value-feedback participants could only infer how much change in force was necessary on the next trial, but not how this should be implemented (increase or decrease in force). Target force levels varied according to a noisy Gaussian decaying random walk with a mean of approximately 20% MVC (Fig 1B). Thus, throughout the task participants had to evaluate the value of their previous actions and adapt their force levels on the next trial accordingly.

### Force production and adaptation in PD patients and healthy controls

Before analyzing the neural correlates of Value-feedback evaluation and action adaptation in the STN, we assessed to what extent the performance of treated PD patients ($n = 16$) was comparable to that of healthy people ($n = 15$). In a first step, we investigated whether PD patients were able to produce forces similarly to healthy controls (HCs). We compared multiple measures of force production including the MVC, mean peak force and its temporal derivative (yank), absolute force exerted, and reaction time. None of these measures differed between groups (force and yank traces are shown in Fig 1C, all statistics are listed in S2 Table) showing that medicated patients can express normal force grips albeit with variability across patients (individual traces are illustrated in S1A and S1B Fig). Next, we tested whether participants were able to adapt their actions according to the feedback they received. We found a strong correlation between actual and target force (average rho = 0.486, $P < 0.001$ in both groups). Similarly, Value-feedback correlated with the corresponding absolute change in force on the next trial (average rho = −0.419) in both groups ($P < 0.001$). This remained significant when dividing trials into attempts with too low versus too high force (all $P$-values $< 0.001$ in both groups) and was consistently observed at the single participant level (the correlation coefficient was negative in all participants and significant in 14 of 15 HC and 13 of 16 PD patients) showing that participants were able to follow task instructions and adapt their force according to the feedback. Patients showed overall poorer task performance compared to HCs regarding the difference between actual and target force resulting in lower average Value-feedback ($t_{29}$ = 3.416, d = 1.23, $P = 0.002$, see Figs 1D and S1C and S2 Table and S1 Text). However, these difficulties did not impact patients' overall ability for force adaptation as indicated by similar force level variability (coefficient of variation, $t_{29}$ = 0.572, d = 0.21, $P = 0.572$) and mean by-trial absolute change in force ($t_{29}$ = 0.346, d = 0.12, $P = 0.732$) between groups (Fig 1D). Thus, the main interest of the current study, i.e., how actions are linked to values and how this is used for action adaptation, were similar in medicated PD patients and healthy people.

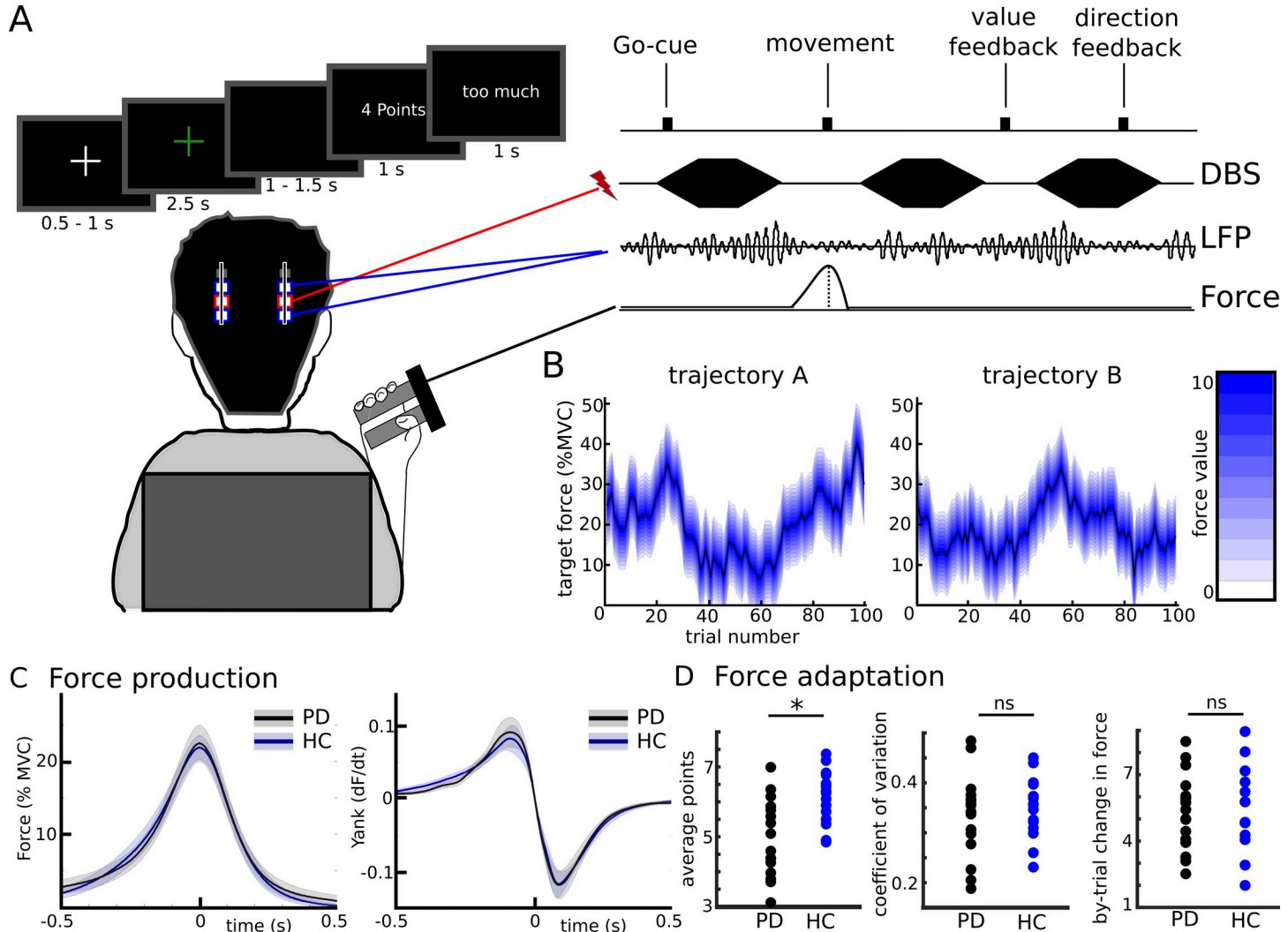

**Fig 1. Paradigm and behavioral results. (A)** After the Go-cue (green fixation cross) participants exerted a certain force to match a target force which had to be inferred based on feedback regarding distance between actual and target force (Value-cue, here 4 out of 10 points) and direction (had the previous force been too little or too much). LFPs were recorded from the bilateral subthalamic nuclei in 2 sessions, together with bursts of DBS in the second session. **(B)** The target force varied according to a Gaussian decaying random walk. The black line indicates the target force and the blue area the reduction of Value depending on the distance from target force. Since patients participated in 2 sessions, 2 trajectories were used. **(C)** Mean force production and yank (first derivative of force) of patients and HCs aligned to peak force (time 0 indicates when peak force was reached). Shaded regions represent SEM. **(D)** Single subject values of different force adaptation measures. By-trial change in force is given as % MVC. DBS, deep brain stimulation; HC, healthy controls; LFP, local field potential; MVC, maximum voluntary contraction; ns, not significant; PD, Parkinson's disease; * indicates a significant difference. Underlying data can be found in Mat1-2 in S1 Data.

### Modulation of STN beta power reflects action evaluation and adaptation

During the task, we recorded local field potentials (LFPs) directly from the STN through temporarily externalized DBS electrodes in PD patients. First, we aligned STN LFPs to the feedback cues (Fig 2A). This showed an increase in STN alpha power from onset of the Value-feedback until 500 ms after the Value-cue (Fig 2B) and a reduction of STN beta power throughout the feedback period (Fig 2C). To assess whether this reflected task-relevant information, we conducted single trial multiple regression analyses using Value-feedback (ranging from 0 to 10) and Direction-feedback (ranging from −10, i.e., >10% MVC too little force, to +10, i.e., >10% MVC too much force) as predictors and STN beta power as dependent variable

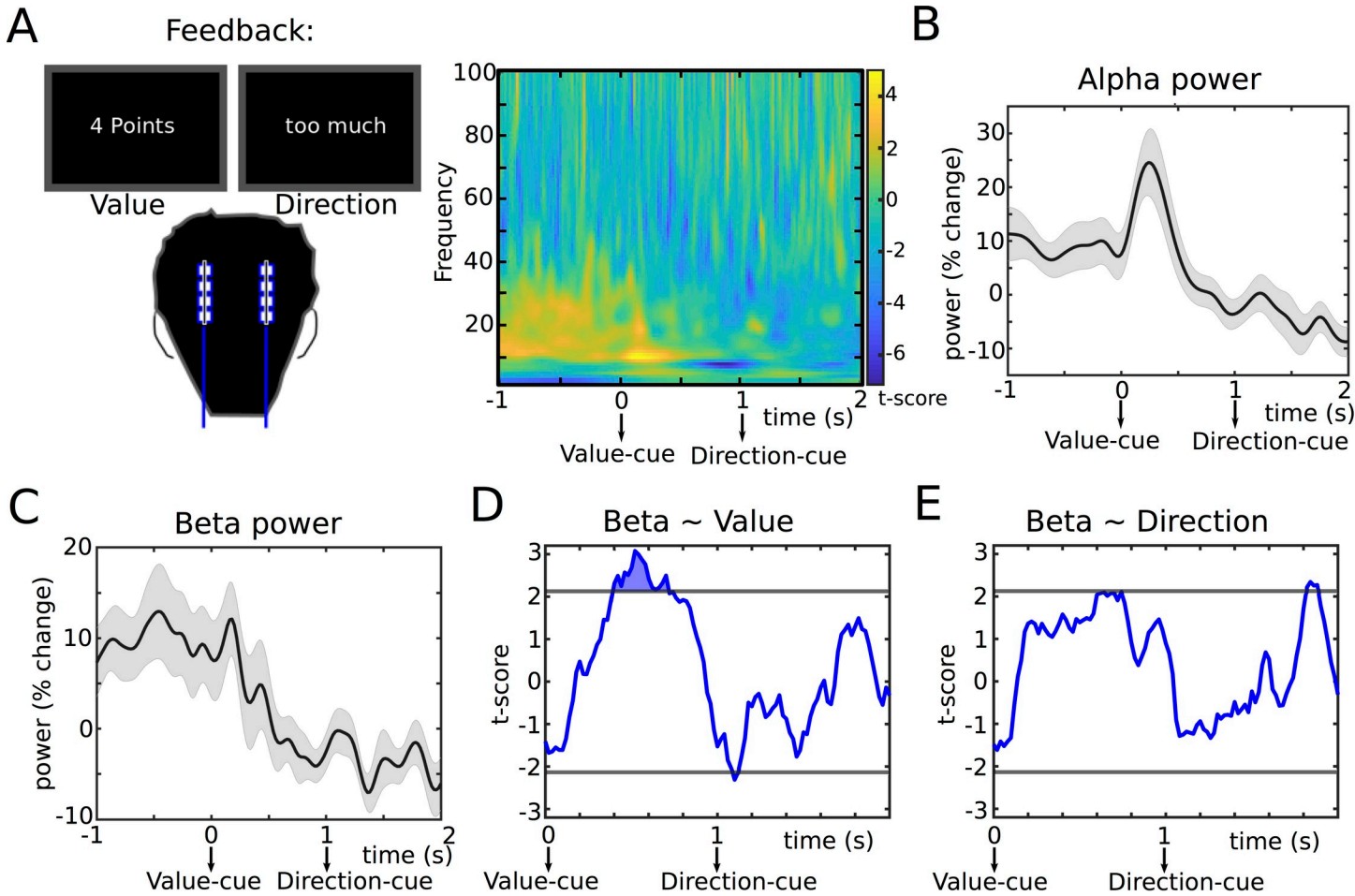

**Fig 2. Correlates of action evaluation. (A)** STN time-frequency spectra aligned to the feedback shown as t-scores (mean across participants divided by standard error). **(B)** Group-average of STN alpha (8–12 Hz) power. There was a visible increase from approximately 0–0.5 s after the Value-feedback. **(C)** Group-average of STN beta power (approximately 13–30 Hz), which was reduced during the feedback period compared to the pre-cue period. **(D)** T-statistic (group level) from the regression between single trial measures of Value-feedback (ranging from 0 to 10) and beta power showing a significant relationship in a time window from 400 to 700 ms after the Value-cue. **(E)** Same as D for regression between Direction-feedback (ranging from −10 to +10) and beta power. Note that effects of Value and Direction were tested in the same multiple regression model. In B and C, shaded areas represent SEM. In D and E, horizontal gray lines show the cluster-building threshold and filled areas indicate significant clusters. Underlying data can be found in Mat3-7 in S1 Data. STN, subthalamic nucleus.

using a sliding window approach (see Methods for details). This revealed a significant, positive relationship between STN beta power and Value from 400 to 700 ms after the Value-cue ($P_{cluster}$ < 0.05, Fig 2D), while there was no significant relationship with Direction (Fig 2E). Thus, STN activity during feedback reflected the absolute difference in force (i.e., Value-feedback) with lower STN beta power being related to lower Value-feedback, but not whether the force had been too high or too low.

To assess the robustness of this result, we conducted several post hoc tests. First, we conducted model comparison. A regression model only containing Value-feedback as independent variable predicted STN activity from the significant time window better than a model only containing Direction-feedback and even the model containing both predictors according to the Bayesian information criterion (BIC of Value model: 2099, Direction model: 2111, combined model: 2103; lower values indicate better performance given model complexity). Second, we excluded each individual participant at a time (while the others remained in the multiple regression model), which showed that the effect of Value-feedback remained highly significant

throughout (mean *P*-value = 0.0007, weakest *P*-value = 0.003 when excluding patient 4). Third, instead of conducting hierarchical regression (see Methods), we applied separate regression analyses for each individual participant and then performed a *t* test on the r-to-z transformed regression slopes, again confirming the significant effect of Value-feedback ($t_{15}$ = 2.873, d = 0.72, *P* = 0.016; 13 of 16 slopes were positive). Furthermore, this effect also remained significant when using an alternative baseline period (500 ms before Value-feedback until Value-feedback, t = 3.458, *P* < 0.001) and when using a "wide" bipolar montage as in the stimulation session (t = 2.490, *P* = 0.013, see below and Methods for details regarding the montage). Finally, excluding individual hemispheres in which neither of the contacts of the bipolar montage localized to an anatomical STN mask according to lead reconstruction (*n* = 4 hemispheres, see Methods, S1 Fig and S1 Table) did not alter significance of the results (t = 3.206, *P* = 0.001).

We also analyzed whether the increase in STN alpha power (see above) reflected the content of the feedback by conducting an additional multiple regression analysis using mean alpha power from onset of the Value-feedback until 500 ms after the Value-cue as dependent variable. This analysis did not show a significant relationship with Value (t = 0.726, *P* = 0.468) or Direction (t = 0.948, *P* = 0.343) neither when using a general definition of alpha (8 to 12 Hz), nor when using participant-specific definitions of alpha (Value: t = 0.658, *P* = 0.511; Direction: t = 0.332, *P* = 0.739). Furthermore, comparing alpha power from this time window between the bipolar contact showing the strongest beta power modulation and the neighboring more ventral contact did not show a significant difference between these different localizations ($t_{13}$ = −1.405, d = −0.351, *P* = 0.184).

Next, we asked whether modulation of STN beta activity also reflected action adaptation. To this end, we aligned STN LFPs to the movement (Fig 3A). This showed an increase in gamma power from −300 ms until peak force (Fig 3B) and a reduction in beta power, which reached its trough around the peak force and increased again after the movement (Fig 3C). To assess whether this was related to participants' behavior, we conducted the same sliding-window multiple regression analysis as above, now using absolute change in force (lower bound at 0 corresponding to no change) and change in force (positive values indicating an increase in force, negative values a decrease in force) as predictors and STN beta power as dependent variable. We found a significant, negative relationship between STN beta power and absolute change in force from 460 to 300 ms before peak force ($P_{cluster}$ < 0.05, Fig 3D). The lower beta power the stronger the absolute change in force. There was no significant relationship with change in force (Fig 3E), i.e., STN beta power reflected how much participants adapted their force, but not whether it increased or decreased.

We conducted several post hoc tests analogously to the analyses during the feedback period. A regression model only containing absolute change in force as independent variable predicted STN activity from the significant time window better than a model only containing change in force and even the model containing both predictors (BIC of absolute change in force model: 2523, change in force model: 2531, combined model: 2530). The effect of absolute change in force remained significant when excluding each individual participant at a time while the others remained in the multiple regression model (mean *P*-value = 0.008, weakest *P*-value = 0.047 when excluding patient 14), when applying separate regression analyses for each individual participant ($t_{15}$ = −2.639, d = 0.66, *P* = 0.019; 12 of 16 slopes were negative), when using an alternative baseline period (500 ms before Value-feedback until Value-feedback, t = −2.989, *P* = 0.003), when using a "wide" bipolar montage (t = −2.286, *P* = 0.022), and when excluding 4 individual hemispheres in which contacts were localized outside an anatomical STN mask according to lead reconstruction (t = −2.743, *P* = 0.006). As expected, the relationship between STN beta power and absolute change in force was not present when aligning

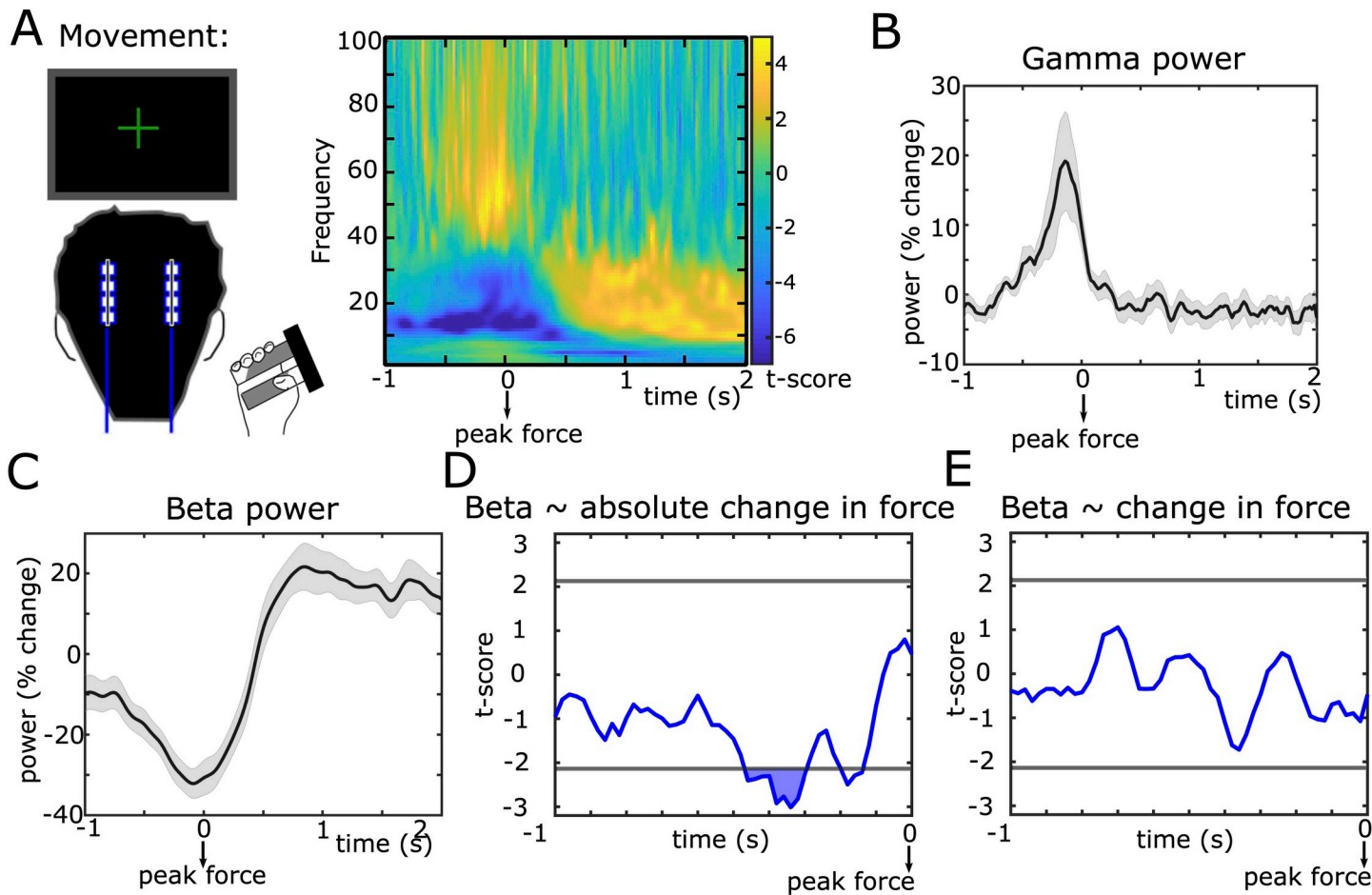

**Fig 3. Correlates of action adaptation. (A)** STN time-frequency spectra aligned to peak force feedback shown as t-scores (mean across participants divided by standard error). **(B)** Group-average of STN gamma (55–80 Hz) power, which increased from approximately 0.3 before peak force until peak force. **(C)** Group-average of STN beta power, which decreased during the movement reaching the trough roughly at the peak force. **(D)** T-statistic (group level) from the regression between single trial measures of absolute change in force and beta power showing a significant relationship in a time window from 460 to 300 ms before peak-force. **(E)** Same as D for regression between change in force (positive values if force was higher, negative values if force was lower compared to previous trial) and beta power. Note that effects of absolute change in force and change in force were tested in the same multiple regression model. In B and C, shaded areas represent SEM. In D and E, horizontal gray lines show the cluster-building threshold and filled areas indicate significant clusters. Underlying data can be found in Mat8-12 in S1 Data. STN, subthalamic nucleus.

STN LFPs to the following Value-feedback (S3 Fig), i.e., it was locked to movement execution, not prior to feedback.

We also analyzed whether the increase in gamma power was related to force adaptation by conducting an additional multiple regression analysis using mean gamma power from −300 ms until peak force as dependent variable. This did not reveal a significant relationship between gamma power and absolute change in force (t = −0.760, P = 0.448) or change in force (t = 0.079, P = 0.937) using a general definition of gamma (55 to 80 Hz), nor when using participant-specific definitions of gamma based on the movement-related gamma increase (absolute change in force: t = −1.069, P = 0.285; change in force: t = −0.390, P = 0.696). Furthermore, comparing gamma power from this time window between the bipolar contact showing the strongest beta power modulation and the neighboring more ventral contact did not show a significant difference (t$_{13}$ = −1.594, d = −0.530, P = 0.135).

Since Value-feedback and action adaptation were correlated (lower Value-feedback resulted in stronger absolute changes in force), it could be that STN beta power after the Value-cue was

primarily related to action adaptation and that this might drive the correlation with Value-feedback. However, a control regression analysis between absolute change in force on the next trial and STN beta power 400 to 700 ms after the Value-cue (see above) did not show a significant effect (t = −0.357, $P$ = 0.721). While this suggests that STN beta power during the feedback was most closely related to the content of this feedback, it does not necessarily entail that it was not relevant for action adaptation since the Value-feedback was informative of the necessary behavioral adaptations. To directly test whether STN activity played a causal role in action adaptation, participants performed the same task in a second session, where bursts of electrical stimulation were applied to the STN. Short bursts (mean duration: 250 ms) were given randomly throughout the task so that in any given 100 ms time window stimulation was applied in approximately 50% of trials (S4A–S4C Fig). This allowed us to compare timing-specific effects of STN stimulation on action adaptation without having to focus on any a priori-defined time windows [25]. Based on the LFP regression analyses, we hypothesized that STN stimulation during the feedback and movement period should affect how much participants adapted their force.

## STN causally contributes to action adaptation

First, we aligned the data to the feedback (Fig 4A). Comparing trials in which stimulation had been applied to trials without stimulation in a sliding-window approach (see Methods for more details), we found a significant, positive effect of STN stimulation on absolute change in force in a restricted time window from 180 to 340 ms after the Value-feedback ($P_{cluster} < 0.05$, see Fig 4B and 4C), hereafter termed $DBS_{value}$. Here, stimulation led to a stronger absolute change in force on the next trial. Since there appeared to be additional, albeit weaker, effects later during the trial (at approximately 600 and approximately 1,000 ms, see Fig 4B), we also directly compared $DBS_{value}$ to these time windows showing that the cluster at $DBS_{value}$ was significantly stronger than both later clusters (respectively, t = 3.559, d = 1.30, $P_{corrected}$ = 0.003 and t = 3.580, d = 1.52, $P_{corrected}$ = 0.003). The effect of $DBS_{value}$ on the absolute change in force did not differ depending on whether the force had been too high or too low ($t_{13}$ = 0.969, d = 0.39, $P$ = 0.351) nor whether the Value-feedback had been relatively high or low (divided by median split at 5 points; $t_{13}$ = 0.395, d = 0.14, $P$ = 0.699). Further supplemental analyses did not show any effects of stimulation on the change in force (increase versus decrease) or collected points (Value-feedback), see S4D and S4F Fig. Since stimulation was not fixed to certain time windows, but applied randomly throughout trials, it was possible that the behavioral effect of DBS during the feedback period required additional stimulation before the next movement. To test this, we compared effects of $DBS_{value}$ for trials with versus without DBS bursts occurring in the following critical premovement period ($DBS_{move}$, see below). This post hoc analysis showed that the effect of $DBS_{value}$ on absolute change in force was significant both for trials where stimulation was applied ($t_{13}$ = 3.174, d = 0.63, $P_{corrected}$ = 0.015) and trials where stimulation was not applied in the following premovement period ($t_{13}$ = 3.106, d = 0.56, $P_{corrected}$ = 0.017) and that the 2 conditions ($DBS_{value}$ with versus without premovement stimulation) were not significantly different from each other ($t_{13}$ = 0.146, d = 0.05, $P$ = 0.886).

Notably, the $DBS_{value}$ time window immediately preceded the period in which STN beta power reflected Value-feedback (400 to 700 ms after the Value-cue, see Fig 2D). Since previous studies have demonstrated that STN stimulation reduces beta power [26–28], this suggests that stimulation during $DBS_{value}$ might have modulated STN beta activity in this 400 to 700 ms post-cue time window. To test this, we analyzed STN beta power from the contacts surrounding the stimulation electrode during the stimulation session. Using common-mode rejection and artifact correction (see Methods), we were able to recover the normal (i.e., as observed in

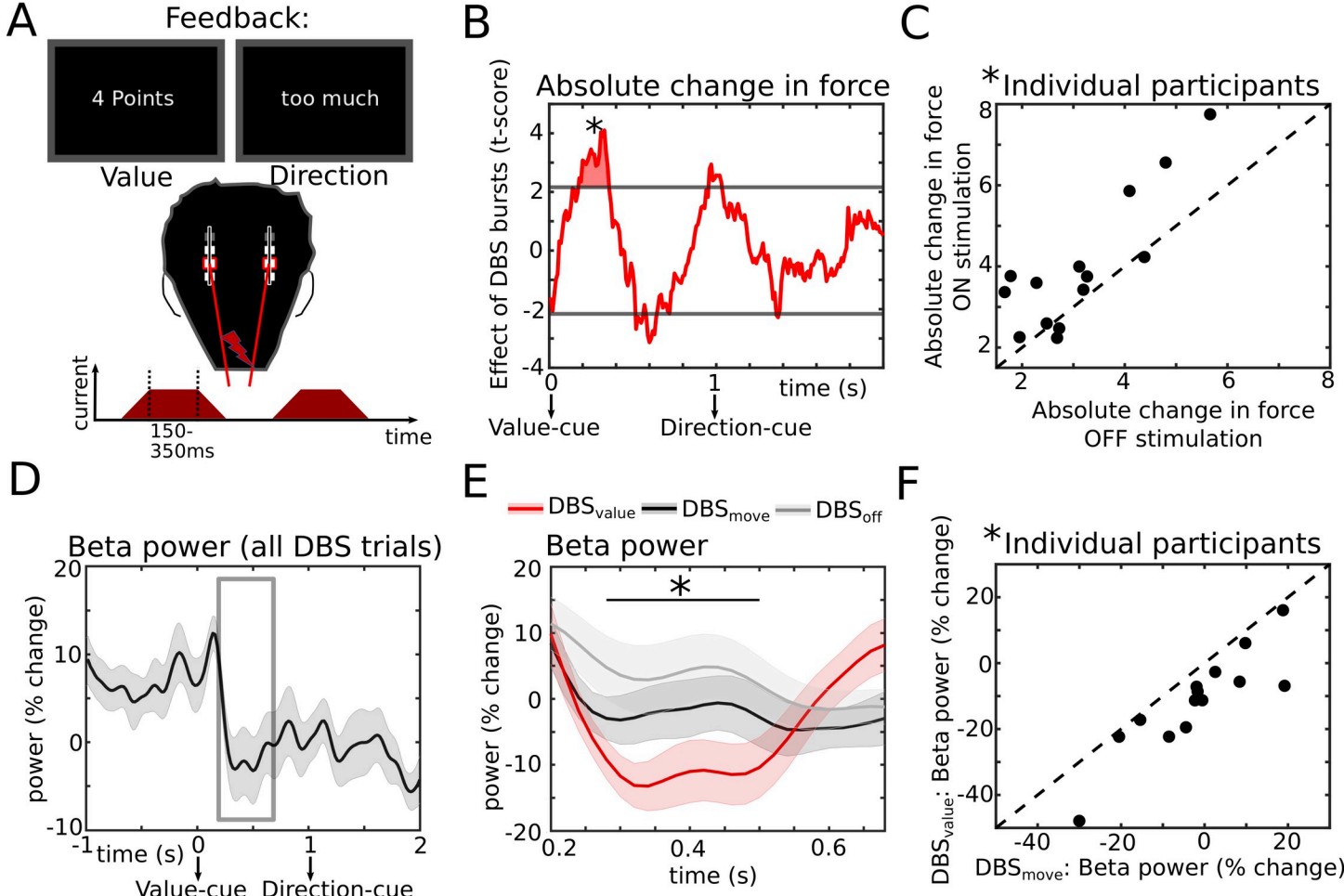

**Fig 4. Causal effects of STN activity on action adaptation during feedback.** (A) Bursts of STN stimulation were applied at random time points throughout the task including the feedback time period (see also Fig 1A). (B) T-statistic (group level) of the effect of STN stimulation, which increased the absolute change in force on the next trial when applied in a time window from 180 to 340 ms (termed DBS$_{value}$) after the Value-cue. Horizontal gray lines show the cluster-building threshold and the filled area indicates the significant cluster. (C) Individual participant data from the significant time window from B (DBS$_{value}$, marked by an "*") for ON vs. OFF stimulation trials. (D) Beta power from the stimulation session aligned to the feedback averaged across participants. The gray rectangle indicates the time window from which beta activity was extracted for evaluating effects of stimulation on beta power. (E) At the group level, stimulation at DBS$_{value}$ (red) reduced beta power compared to control trials in which DBS was applied in a time window from 680 to 580 ms before peak force (termed DBS$_{move}$, black). Beta power from the no-stimulation session (DBS$_{off}$, gray) is also plotted. The black line with an "*" indicates the time window with a significant difference between DBS$_{value}$ and DBS$_{move}$. (F) Individual participant data for the significant time window from E (marked by an "*"). Shaded areas in D and E represent SEM. DBS, deep brain stimulation; STN, subthalamic nucleus. Underlying data can be found in Mat13-17 in S1 Data.

the off stimulation session) feedback-modulation of STN beta power (see Figs 4D and S5). As expected from previous studies [26–28], aligning STN beta power to onset of stimulation bursts showed a marked (approximately 40%) reduction in beta power, while it did not significantly alter power in other frequency bands (S6 Fig and S3 Table). We then extracted beta power from a time window of interest spanning DBS$_{value}$ and the time window where STN beta power reflected Value-feedback (i.e., from 180 to 700 ms after the Value-cue, see gray rectangle in Fig 4D) and compared it to control trials. In these control trials, stimulation also affected participants' behavior but in a different time window (DBS$_{move}$, see below). This analysis showed a significant reduction of beta power by stimulation during DBS$_{value}$ from 280 to 500 ms after the Value-cue (P$_{cluster}$ < 0.05, Fig 4E and 4F, see gray trace for beta power off stimulation). Thus, stimulation reduced STN beta power after the Value-cue, where beta

activity normally reflected the absolute difference in force (i.e., Value-feedback) with lower beta power being related to larger differences necessitating stronger adaptation. Behaviorally, stimulation during DBS$_{value}$ led to an increase in action adaptation as would be expected from trials with lower Value-feedback.

Next, we aligned data to the movement and conducted the equivalent analysis as above (Fig 5A). This revealed a significant, positive effect of STN stimulation on absolute change in force in a distinct time window from 680 to 580 ms before peak force (P$_{cluster}$ < 0.05, Fig 5B and 5C), hereafter termed DBS$_{move}$. Here, stimulation led to a stronger absolute change in force on the current trial. Interestingly, across participants effects of DBS$_{move}$ and effects of DBS during the feedback period (DBS$_{value}$) were not correlated (rho = −0.030, P = 0.920). The effect of DBS$_{move}$ did not significantly differ depending on whether the force on the previous trial had

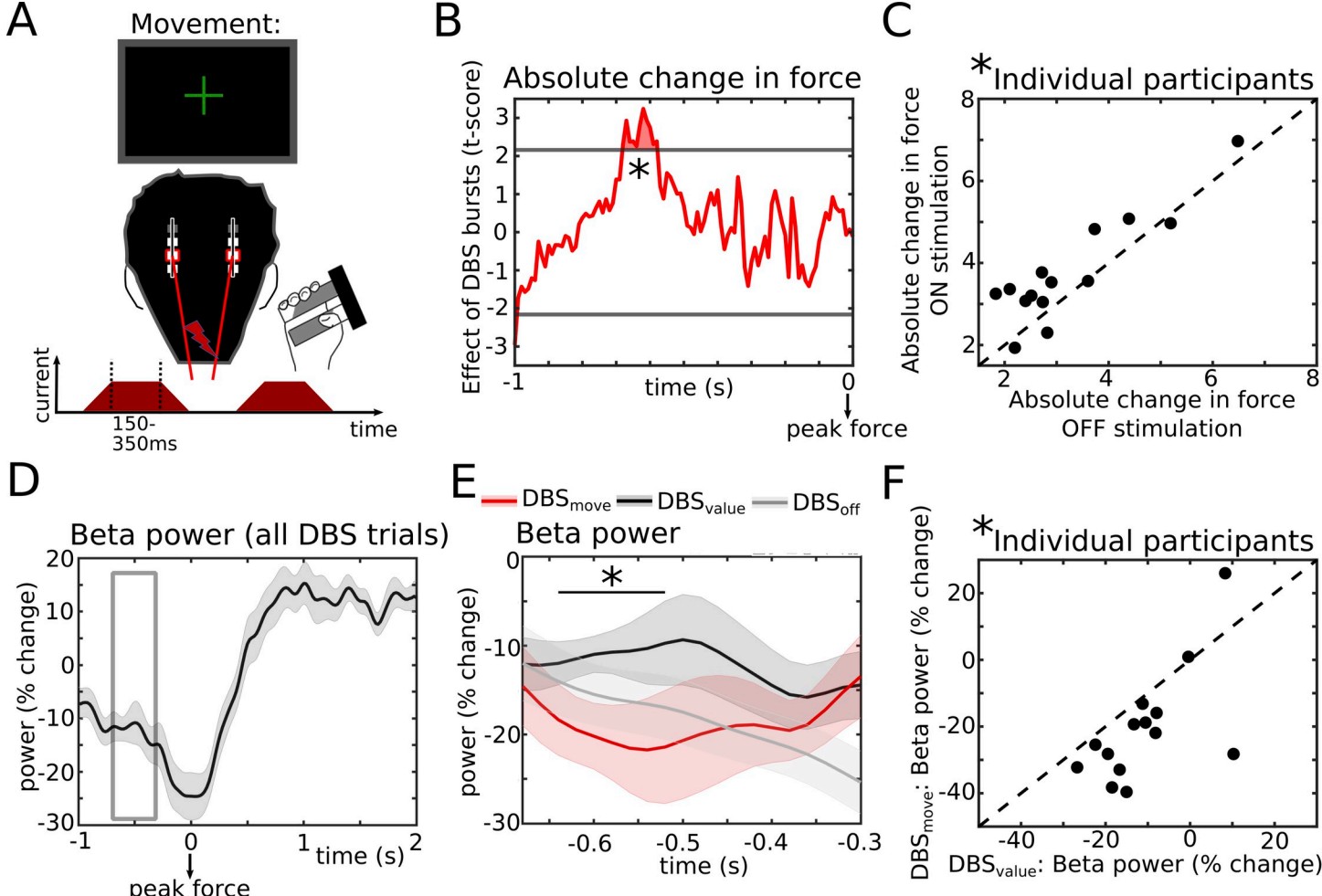

**Fig 5. Causal effects of STN activity on action adaptation in relation to movement.** (A) Bursts of STN stimulation were applied at random time points throughout the task including the movement time period. (B) T-statistic (group level) of the effect of STN stimulation, which increased the absolute change in force on the current trial when applied in a time window from 680 to 580 ms (DBS$_{move}$) before peak force. Horizontal gray lines show the cluster-building threshold and the filled area indicates the significant cluster. (C) Individual participant data from the significant time window from B (DBS$_{move}$, marked by an "*") for ON vs. OFF stimulation trials. (D) Beta power from the stimulation session aligned to the movement averaged across participants. The gray rectangle indicates the time window from which beta activity was extracted for evaluating effects of stimulation on beta power. (E) At the group level, DBS$_{move}$ (red) reduced beta power compared to control trials (DBS$_{value}$, black). Beta power from the no-stimulation session (DBS$_{off}$, gray) is also plotted. The black line with an "*" indicates the time window with a significant difference between DBS$_{move}$ and DBS$_{value}$. (F) Individual participant data for the significant time window from E (marked by an "*"). Shaded areas in D and E represent SEM. DBS, deep brain stimulation; STN, subthalamic nucleus. Underlying data can be found in Mat18-22 in S1 Data.

been too low or too high ($t_{13} = -1.659$, d = $-0.76$, $P = 0.123$) nor whether the previous Value-feedback had been relatively high or low ($t_{13} = -0.412$, d = $-0.17$, $P = 0.687$). Further supplemental analyses did not show any effects of stimulation on the change in force (increase versus decrease) or collected points (Value-feedback), see S4E and S4G Fig.

Of note, DBS$_{move}$ was in close proximity to the period in which STN beta power reflected absolute change in force off stimulation (440 to 300 ms before peak force, see Fig 3D). Analogously to the analysis of DBS$_{value}$, we therefore extracted beta power from the window comprising this window and DBS$_{move}$ (i.e., from 680 to 300 ms before peak force, see gray rectangle in Fig 5D) now comparing DBS$_{move}$ to DBS$_{value}$ as control trials. STN beta power was reduced by stimulation during DBS$_{move}$ up to 340 ms before peak force with a significant cluster from 640 to 520 ms before peak force ($P_{cluster} < 0.05$, Fig 5E and 5F). Thus, here stimulation led to a stronger absolute change in force and reduced STN beta power prior to reaching peak force, close to the time window where lower beta power normally (i.e., off stimulation) reflected this behavioral adaptation.

## Behavioral effects of STN stimulation are related to connectivity to motor cortex

Applying bursts of DBS showed that effects of STN stimulation showed temporal specificity, i.e., only occurred in distinct, short time windows. Did stimulation effects also show spatial specificity? To address this in a final analysis, we took advantage of interindividual variability in the exact placement of DBS electrodes in relation to the STN and interconnected networks (reconstructed individual DBS leads and stimulation contacts are shown in Fig 6A). Using a recently developed DBS connectivity analysis approach [29], we estimated the volume of activated tissue [30] in each hemisphere and used this as seed region for connectivity analysis based on a published connectome dataset in PD patients [31,32]. Then, in a hypothesis-driven approach, we tested whether the DBS effects on force adaptation were related to the extent to which the stimulated volume was connected to motor cortex. We chose the primary motor cortex as region of interest, since it has in previous studies been shown to be involved in force production and adaptation [19,33,34], is structurally connected to STN (in particular to dorsal STN expressing STN beta activity [35]) and related to clinical improvement after STN-DBS in PD [36]. As a control region, we used temporal cortex, since while temporal cortex also is

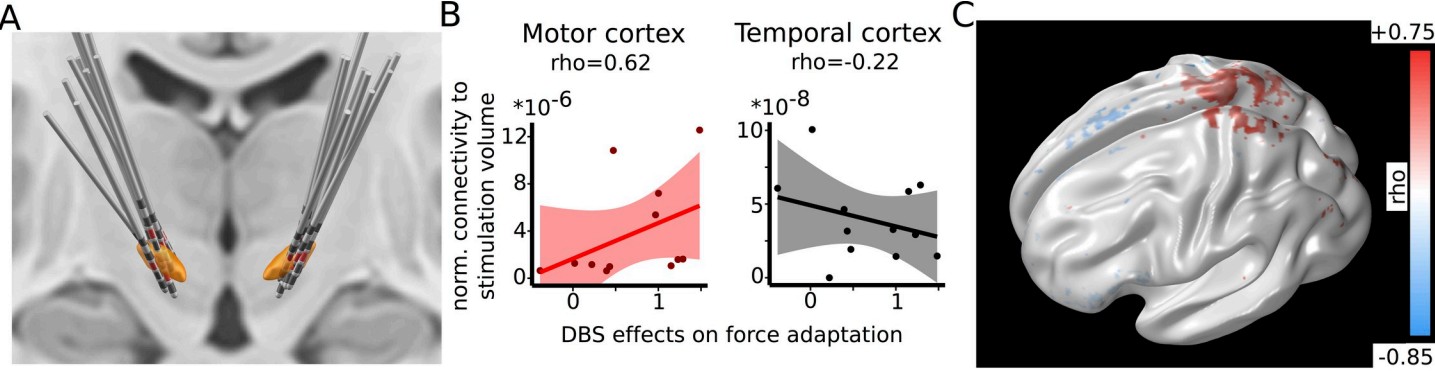

**Fig 6. Connectivity mapping of DBS effects. (A)** 3D view of reconstructed bilateral leads overlaid on an anatomical mask of the STN (orange color). Active contacts, i.e., electrodes used for DBS, are marked in red showing interindividual spatial variability. **(B)** Results from Spearman rank correlations between connectivity of the stimulation volume to motor cortex (left panel) and temporal cortex (control site, right panel) with interindividual differences in the behavioral effect of DBS on absolute change in force (mean of DBS$_{value}$ and DBS$_{move}$, see Methods). **(C)** For illustration purposes, this 3D map of the cerebral cortex shows voxel-wise correlation values between cortical connectivity to stimulation volumes and the behavioral effect of DBS (same variable as in B). Positive values (red color) largely cluster around motor cortical areas. The figure is thresholded at rho = ± 0.1 for illustration purposes only. DBS, deep brain stimulation; STN, subthalamic nucleus.

structurally connected to the STN [37], it is not primarily involved in force production nor related to clinical improvement in PD. In this analysis, a positive correlation indicates that if connections to the given cortical area were stimulated more strongly due to the individual localization of the stimulation electrode and stimulation parameters, then the behavioral DBS effect on absolute change in force in this participant were also stronger. In line with our hypothesis, we found a significant positive correlation between connectivity of the stimulation volume to motor cortex (rho = 0.62, $P_{corrected}$ = 0.037), but not to temporal cortex (rho = −0.22, $P_{corrected}$ = 0.499), see Fig 6B. The motor cortex connectivity correlation was also significantly stronger than the temporal cortex correlation in a direct comparison (z = 2.082, $P$ = 0.037). These results suggest that the behavioral STN DBS effects not only showed temporal, but also spatial specificity for motor networks (illustrated in Fig 6C).

## Discussion

Among the many possibilities afforded by our environment, which actions should we choose and how should we perform them? One solution to this problem is to learn the value of different actions and how these are performed through trial-and-error and compare their outcomes to an expected value. Outcomes surpassing the expected value should be reinforced, while poor outcomes require adaptation. This reinforcement learning framework has proven to be highly useful in predicting behavior and even appears to have direct correlates in cortex–basal ganglia networks [5–9]. At better-than-expected outcomes, striatal dopamine is released from dopaminergic midbrain neurons, while a temporary reduction in dopaminergic firing has often been observed after worse-than-expected outcomes [6–9]. Due to differential modulation of the net-facilitatory direct basal ganglia pathway through D1-type receptors and the net-inhibitory indirect basal ganglia pathway through D2-type receptors, this can modify cortical excitability and shape neural dynamics depending on action–outcome contingencies [3,10–14]. The aim of the current study was to elucidate the electrophysiological correlates of action evaluation and modification in the basal ganglia and assess their causal contributions to behavioral adaptations in humans.

Recording activity directly from the STN, a central part of the net-inhibitory indirect pathway [11], we found that reduced levels of beta power were related to larger absolute deviations between actual and target force as well as between larger absolute adjustments of force irrespective of direction. That is, this relationship did not depend on whether the force had been too low or too high or whether it increased or decreased on the next trial. This is compatible with previous studies suggesting that STN is critically involved in changes of actions or motor states rather than merely kinematics [19,20,38]. Another way to put this is that reduction of STN beta activity may reflect an effort necessary for changing between neural and/or behavioral states [39]. This cost can be disambiguated from mere muscular energy costs since, for example, it increases when an isometric contraction is reduced or terminated [19], or a weaker effector is used [17].

Combining STN stimulation and simultaneous recordings of STN activity, we found that short bursts of STN DBS were sufficient to modify adaptive behavior when applied in critical time windows and their effects on STN beta activity were consistent with their effect on behavior. Reduction of STN beta activity within or close to the time windows where lower levels of beta normally (off stimulation) warranted change led to stronger force adaptation. This demonstrates that precisely delivered electrical stimulation can modify movement adaptation in humans. In the current study, DBS bursts were delivered randomly and, as expected, increasing the absolute change in force through stimulation did not improve task performance as reflected by the number of collected points arguing against general effects of DBS on alertness

or task engagement. It would be interesting to test whether task performance can be improved if stimulation is specifically delivered in trials where change is warranted.

The rapid cycling between stimulation and no stimulation had the advantage of not having to define any a priori temporal windows of interest [25]. However, this design had the shortcoming that there were no trials in which stimulation was not applied at all and allows for the possibility that the key timings for stimulation effects were conditional on stimulation also occurring somewhere else in each trial. The fact remains, however, that stimulation in the identified time windows was still necessary for behavioral impacts. Moreover, we accounted for potential conditionality by conducting additional control analyses demonstrating that the behavioral effect of DBS during Value-feedback did not depend on premovement stimulation (see above). Future studies could help further exclude the possibility of conditionality by limiting stimulation to predefined timings in each trial.

Since it has been proposed that dopamine release is related to reductions in beta activity of the STN [40] and some studies have observed a temporary dip in dopaminergic neuron firing after worse-than-expected outcomes [8,9], this would predict that lower action values in our study should lead to increases in beta activity. However, we observed the opposite namely a decrease in beta activity after worse outcomes and stronger action adaptation in line with previous studies [20, 21]. This suggests that there is not a simple linear negative relationship, but a more intricate relationship between dopamine release and beta power at finer timescales [41]. It should also be noted that STN beta activity mainly localizes to subparts of the STN connected to cortical motor areas [42,43] rather than "reward"-related networks, which might provide distinct computations [44–46], i.e., dopamine release in "motor" basal ganglia circuits might not directly correspond to a value or prediction error-like signal. Electrode reconstruction confirmed that the contacts used for LFP recordings mainly localized to the dorsolateral STN. Furthermore, correlating interindividual differences in the cortical connectivity profile of STN stimulation volumes and differences in the DBS-related behavioral effects on force adaptation showed a significant correlation with motor cortex connectivity, while a control site (temporal cortex) did not show this relationship. This lends further support for a critical role of STN and interconnected cortical motor areas in force production and adaptation [16–21,33,34,47,48] and, similarly to the temporal specificity of DBS, argues against nonspecific, general stimulation effects underlying the behavioral changes.

We observed an increase in alpha activity after the Value-feedback, which however was not correlated to Value, i.e., it increased irrespective of negative or positive outcomes. Alpha activity in STN has mainly been related to attentional mechanisms, since it increases after salient stimuli [49] and is at rest coherent with temporoparietal cortex [50–52], suggesting that it reflects a response to a salient cue rather than its content. This contrasts with STN beta power, which scaled with the content of the Value-feedback, arguing against a strong effect of feedback cue order on STN beta power that would predict a general change in beta power similar to what we observed for STN alpha power. Since the duration of Value-feedback was fixed, the increase in alpha activity could in theory also reflect anticipation of the Direction-feedback onset. To what extent feedback cue order and cue anticipation affects STN activity in distinct frequency bands could be tested in future studies by switching the order of Value- and Direction-feedback and varying feedback durations.

In addition, we found a strong increase in gamma activity, which was maximal around the peak force. While gamma activity is closely related to force levels for stronger forces [15,16,49] and movement velocity [49,53,54], it appears not to be related to action adaptation, at least in the current paradigm. While these findings demonstrate the relative specificity of a component of STN beta activity as a signal reflecting action adaptation, we do not claim that complex behavior like action adaptation can be reduced to one simple LFP signal but rather will be

reflected by multidimensional population dynamics across neural networks [55]. This does not only entail cortical and basal ganglia areas outside the STN, but also distinct subcortical nuclei such as the habenula or rostro-medial tegmental nucleus [56,57], which we could not investigate in the current study.

Are the mechanisms that we studied here relevant in the healthy state? PD patients can express abnormally slow movements (especially during large amplitude movements and large forces [58–60]), in particular OFF medication. To address this, we carefully designed the task using relatively low forces and limited cognitive demands, and assessed all patients in their ON medication state when clinical impairment is less pronounced. Despite this, patients had difficulties in precisely producing lower forces, which might be related to impaired dexterity [61,62] or the necessity to grip the dynamometer with a certain baseline force due to tremor (see S1 Text). However, patients had overall similar kinetics and, more importantly, similar measures of force adaptation as healthy people, which was the main interest of this study.

In some patients, dopaminergic medication can lead to abnormal reward processing mechanisms and the development of impulse control disorders. While this study did not include patients suffering from impulse control disorders, we cannot with certainty discount that medication might in some patients have negatively affected their ability to adapt behavior. To assess this, studies testing patients both ON and OFF dopaminergic medication could be conducted.

Another issue with the present investigation is that PD patients have reduced levels of dopamine and exaggerated STN beta activity when OFF medication [40], which might limit the generalizability of STN activity modulations during the task. As detailed above, all patients were studied ON medication, which reduces resting STN beta power [40] and leads to more pronounced modulation of beta power related to cues and movement compared to OFF medication [63,64]. Furthermore, there is a vast literature beyond STN LFP recordings in PD patients demonstrating neural correlates of force production in the basal ganglia including non-invasive recordings with functional magnetic resonance imaging in healthy people [65,66], invasive electrophysiological recordings in healthy nonhuman primates [67,68], and invasive electrophysiological recordings in humans with neurological disorders other than PD [69]. In addition, cortical beta oscillations have been shown to reflect action adaptation in healthy humans using electroencephalography [47,70]. Together, this suggests that our findings may generalize beyond the studied patient group.

In summary, we here demonstrate that action evaluation and adaptation are reflected by dynamic STN beta activity and that causal manipulation of such activity can modify action adaptation in humans. More broadly, our results are in line with dynamic reductions in beta activity underlying adaptive processing. Future studies are warranted to assess whether this can be leveraged to reestablish physiological processing and assist motor execution and even learning [71] in patients suffering from neurological disorders.

## Methods

### Ethics statement

All participants gave written informed consent to participate in the study, which was approved by the local ethics committee (State Medical Association of Rhineland-Palatinate) and conducted in accordance with the declaration of Helsinki.

### Participants

We recruited 16 patients with PD, who had undergone STN DBS surgery prior to the experimental recordings at University Medical Center at the Johannes Gutenberg University Mainz,

Germany. Clinical details are listed in S1 Table. Lead localization was verified by microelectrode recordings, monitoring the clinical effect and side effects during operation, as well as through postoperative stereotactic computerized topography (CT), see S2 Fig. Bilateral STN LFP were recorded and DBS applied through externalized electrode extension cables. The experiment was conducted in the immediate postoperative period 1 to 3 days after insertion of the DBS lead (Abbott 6170), before implantation of the subcutaneous pulse generator, in the ON medication state (i.e., patients' usual antiparkinsonian medication). While externalized DBS leads provide the unique opportunity to directly record LFPs from the STN and apply specific stimulation pulses, a putative downside is that LFPs might be reduced due to the stun effect [72]. Each DBS lead had 8 contacts on 4 vertical levels. The uppermost and lowermost contacts are ring contacts while both middle contacts consist of 3 directional contacts. The height of each contact is 1.5 mm and there is a 0.5 mm space in between the 4 vertical levels (see S7A and S7B Fig). As a control group, we enrolled 16 HC participants without any neurological or psychiatric conditions. The groups did not differ regarding age (PD: 66 ± 13 years; HC: 67 ± 8 years, mean ± standard deviation; $t_{30} = -0.227$, $P = 0.822$, d = 0.08, independent samples $t$ test), handedness (1 left-handed person in each group as revealed by self-report, $P = 1$, Fischer's exact test), or sex (14 male in PD group, 11 male in HC group, $P = 0.394$, Fischer's exact test). Two of the included PD patients (PD04 and PD07) did not participate in the second session with STN burst stimulation (see below) due to fatigue. One healthy participant had to be excluded due to miscalibration of the force device.

## Experimental task

We designed the experimental task so that participants had to constantly adapt the force they applied, while avoiding forces close to their maximum, or forces close to 0. Furthermore, the outcome should not be unambiguously predictable, so that participants had to wait for the feedback-cues after the movement, while also not being purely random. To this end, we computed trajectories of target force levels that noisily varied around their mean based on a decaying Gaussian random walk, which has been applied for studying decision-making under uncertainty [24]. In particular, the target force μ at each trial t was given by the following:

$$\mu_{t+1} = \mu_t \lambda + (1 - \lambda) * \theta + \nu. \tag{1}$$

The start-value $\mu_1$ was drawn from a Gaussian distribution with a mean of 25 (% MVC) and a standard deviation of 2, while ν was a noise term drawn from a zero-mean Gaussian with a standard deviation of 2, and λ and θ were constants describing the rate of decay and the value towards which $\mu_t$ decayed to and were set to, respectively, 0.98 and 0.25 [24]. The 2 trajectories (A and B, see Fig 1B) used in this study were derived from simulations with the above given equations and kept constant across participants to facilitate comparisons between subjects. The resulting mean target force over trials was approximately 20% MVC in both trajectories (i.e., slightly lower than θ). We used 2 trajectories for the target force levels, because patients participated in 2 sessions (1 without stimulation and 1 with bursts of STN stimulation, see below). Since healthy participants only participated in 1 session (of note only the first session was compared between groups, i.e., PD patients off stimulation versus HC), we counterbalanced the order of trajectories between session A and session B for PD patients and matched trajectories between patients and HC.

MVC was calculated as the median out of 3 attempts to press a manual dynamometer as hard as possible. For these attempts and all movements throughout the task, participants were instructed to apply relatively short force grips (in contrast to long-lasting isometric contractions) to avoid fatigue.

After estimating the MVC, participants performed a short training session in which they could get accustomed to the dynamometer and the initial force level. For this session, the participants were told that the target force would be 25% of their maximal force, which was kept constant across the 10 training trials.

Next, the experimental session was conducted. The participants' goal was to collect a maximum number of points by exerting forces that were as close to the target force level as possible at each trial. Participants were aware that the initial level was close to the training session, but that this would change over time. Since the target force was not shown on the screen, participants had to infer this based on the feedback they received. At the beginning of each trial, a white fixation cross was shown for an average duration of 0.75 s (randomly jittered between 0.5 and 1 s). When the fixation cross turned green, participants were instructed to produce the force they predicted to be as close as possible to the target force. The movement was allowed any time in a 2.5 s window and the fixation cross remained green for this whole duration irrespective of the timing of the movement. Participants were discouraged from exerting multiple presses even if they perceived that their first press was suboptimal and to refrain from any movement after the first press and wait for the feedback. After this 2.5 s window, a black screen was shown for 1 to 1.5 s after which the Value-feedback was presented for 1 s. Values ranged from 0 (worst) to 10 points (best) and depended on the linear distance of the actual force from the target force, i.e., 0% to 1% MVC difference resulted in 10 points, 1% to 2% MVC difference in 9 points, etc. Any difference >10% MVC resulted in 0 points. After this Value-feedback, the Direction-feedback was shown for 1 s indicating whether the actual force had been "too much" (German: "zu viel") or "too little" (German: "zu wenig"), after which the next trial began with a white fixation cross. The experimental session comprised 100 trials corresponding to approximately 10 min. At the end of the session, the sum of collected points was shown.

All cues were presented on a MacBook Pro (MacOS Mojave, version 10.14.6, 13.3 inch display, 60 Hz refresh rate) using PsychoPy v1.8 [73] implemented in Python 2. The display was viewed from a comfortable distance of approximately 50 cm. Hand grip force was measured with a dynamometer (MIE Medical Research, Leeds, United Kingdom), which the participants held in their dominant hand with their forearm comfortably positioned on the armrest of the chair. Two people in each group used their nondominant left hand, because of discomfort on the right side. The analogue force measurements were analogue-to-digital converted and sent to the PsychoPy software through a labjack u3 system (Labjack Corporation, Lakewood, Colorado, United States of America) as well as to the LFP recording device. In PsychoPy, force was converted to % MVC for each individual participant. Task events were synchronized with the analogue force and LFP recordings (as well as DBS bursts in the second experiment, see below) by a TTL pulse that was sent from Psychopy to the recording software through the labjack system.

### Analysis of behavioral data

All trials without responses or with more than 1 response were excluded. The remaining trials were compared between groups across multiple measures of force production and adaptation to assess the generalizability of the results.

### Force production

For each participant, we calculated the mean peak force (peak force minus baseline), mean peak yank (first derivative of force), mean peak negative yank, and area under the curve (AUC, area between exerted force and baseline) and compared these variables between groups using independent samples $t$ tests. The baseline was computed as median of a 5 s window centered

on the Go-cue (i.e., 2.5 s before until 2.5 s after onset of the green fixation cross) at each trial to account for putative baseline drifts, which was visually inspected at each trial. We also compared the reaction time (mean time from Go-cue to peak force), the MVC (median of 3 attempts, see above) as well as the peak yank-to-peak force slope (Fisher z-transform of Pearson correlation coefficient) between groups.

## Force adaptation

Root mean squared error (RMSE), average Value-feedback (which is closely related to RMSE, since larger errors will results in fewer points collected), mean force error (mean difference in actual versus target force, which reflects whether on average too much or too little force was applied), mean actual force at low versus high target force levels (after median split of target force levels), coefficient of variation (CV, standard deviation of force divided by mean force), and mean by-trial absolute change in force (how much did participants on average change their force from trial to trial) were calculated and compared between groups using independent samples *t* tests. To assess whether participants in general were able to follow task instructions, we also calculated Pearson correlations between actual and target force (successful performance would predict a positive correlation) as well as Value-feedback and absolute change in force on the next trial (successful performance would predict a negative correlation). For all measures that reflected between-trial adaptation, only trials where the previous trial also had been valid (i.e., 2 consecutive valid trials) were included. Throughout the analysis, all data were tested for normality using Lilliefors test before conducting parametric tests. Effect sizes were computed using Cohen's d (Matlab function *computeCohen_d*). All results are listed in S2 Table.

## Processing of STN LFPs

LFPs were sampled from bilateral STN at 2,048 Hz, bandpass filtered between 0.5 and 500 Hz and amplified with a TMSi porti device (TMS International, Enschede, the Netherlands). The same system was used for recording the force measures and TTL pulses (see above) through auxiliary input channels. The whole recording was visually inspected for artifacts off-line in Spike2 (Cambridge Electronic Design, Cambridge, UK) and noisy trials were rejected. After artifact rejection (on behavioral and neurophysiological grounds), approximately 78 trials per patient and 1,246 trials in total remained. Further analysis of the data was performed using FieldTrip [74] implemented in Matlab (R 2019a, The MathWorks, Natick, Massachusetts, USA). All scripts are available on https://data.mrc.ox.ac.uk [75]. The data were imported to Matlab, high-pass filtered at 1 Hz using a fourth-order Butterworth filter, bandstop filtered between 49 and 51 Hz (FieldTrip function *ft_preprocessing*) and downsampled to 200 Hz using an anti-aliasing filter at 100 Hz (*ft_resample*). A bipolar montage was created from the monopolar recordings by computing the difference between the most dorsal omnidirectional contact and the neighboring 3 dorsal directional contacts, between the 3 dorsal and corresponding 3 ventral directional contacts, as well as the 3 ventral directional contacts and neighboring most ventral omnidirectional contact resulting in 9 bipolar channels per STN (*ft_apply_montage*), see S7B Fig. For each bipolar channel, the data were transformed to the frequency domain using the continuous Morlet wavelet transform (width = 7, *ft_freqanalysis*) for frequencies from 2 to 100 Hz using steps of 1 Hz and 20 ms throughout the whole recording. Power of each frequency was baseline corrected (*ft_freqbaseline*) relative to the mean power of that frequency across the whole recording [25,76] excluding time periods with large artifacts. The resulting spectra were epoched and aligned with, respectively, peak force and feedback-cues (*ft_redefinetrial*). In order to identify the bipolar contact, which showed the strongest

task-related modulation, we analyzed all contacts with respect to their changes in movement-related beta (frequency of maximal modulation defined individually between 13 and 30 Hz) activity. We chose this as a functional localizer, because STN beta activity is localized within the dorsal STN and correlates with motor performance [42,43,63,77], and movement-related beta power modulation was present (defined as >15% reduction during movement) in all hemispheres. For each hemisphere, the contact with the strongest movement-related decrease in beta power was chosen and these contacts then averaged across hemispheres resulting in 1 STN channel per patient. We confirmed the validity of this functional localizer approach by conducting lead localization analysis (see below). When analyzing other frequency bands, we applied commonly used definitions: theta from 4 to 8 Hz, alpha from 8 to 12 Hz, and gamma from 55 to 80 Hz. For control regression analyses (see main text and below), we also used participant-specific definitions of alpha (based on the feedback-related alpha increase) and gamma (based on the movement-related gamma increase) after visual inspection of the individual spectra without applying any predefined thresholds.

## Electrode localization

Electrode localization was carried out using the Lead-DBS toolbox (v.2.5.2; https://www.lead-dbs.org/) with default parameters as described elsewhere [78]. Briefly, using Advanced Normalization Tools (ANTs) preoperative magnetic resonance imaging and postoperative CT scans were corrected for low-frequency intensity non-uniformity with the N4Bias-Field-Correction algorithm, co-registered using a linear transform and normalized into Montreal Neurological Institute (MNI) space (2009b nonlinear asymmetric). Brain shifts in postoperative acquisitions were corrected by applying the "subcortical refine" setting as implemented in Lead-DBS [79]. The reconstructed electrodes (respectively marked at contacts, which were used for LFP recordings and stimulation) were then overlaid on a mask of the STN using the DISTAL atlas [31] to confirm proper targeting, see Figs 6A and S2. In 4 of 26 hemispheres (imaging data was not available in 3 patients), neither of the 2 contacts used for the LFP analysis (see above) were within the STN mask according to lead reconstruction. To assess whether this affected the LFP regression analysis (described below), we excluded hemispheres where the contacts were outside the STN from the LFP regression analysis (i.e., instead of the mean signal across hemispheres, only the LFP from the hemisphere with electrode localization within the STN was used). Of note, each patient had a bipolar contact within the STN in at least 1 hemisphere, i.e., no participants had to be excluded for this control analysis.

## Burst stimulation

After the first session, patients had a short break of approximately 30 to 60 min. During this, the LFPs recorded from bilateral STN were processed and analyzed as described above, but instead of constructing bipolar channels from neighboring electrodes, 2 wider bipolar contacts were constructed to allow recording during stimulation of an intervening contact (S7C Fig). First, directional contacts were averaged to form an omnidirectional contact (resulting in 4 omnidirectional contacts per STN). Then, a dorsal bipolar contact between the most dorsal and second most ventral contact and a ventral bipolar contact between the most ventral and the second most dorsal contact were created. This was done to compute the bipolar contact with the clearest movement modulation of beta activity, since this has been related to localization within or close to the dorsal STN and correlates with motor performance [42,43,63,77] and allows stimulation of the contact in between this bipolar pair to mitigate the stimulation artifact using common mode rejection [25,80,81]. The 2 bipolar contacts on each side were then compared regarding the extent of movement-related beta power modulation and the best

contacts (i.e., with the clearest modulation) chosen as recording electrodes using the electrode in between as active contact for stimulation. DBS was applied using a custom-built device previously validated [80] in pseudo-monopolar mode using reference pads on the patients' shoulders as anode. Frequency (130 Hz) and pulse width (60 μs) were fixed. DBS was not applied continuously, but in short bursts to allow inference on timing-specific effects of stimulation. In more detail, mean DBS burst duration was 250 ms (drawn randomly from a uniform distribution between 150 and 350 ms) and mean burst interval was 150 ms (drawn randomly from a uniform distribution between 75 and 225 ms), which together with ramping up and down of stimulation resulted in a pause between DBS bursts of on average approximately 500 ms (see below, S4A Fig and S1 Table). These parameters were defined based on our previous study of closed-loop DBS [80] and were in simulations shown to result in DBS bursts occurring in approximately 50% of trials in any given 100 ms time window during the experimental task allowing us to compare timing-specific behavioral effects of DBS versus no-DBS. Thus, stimulation in specific time windows could be compared to trials from the same session where no stimulation was applied at the respective time window. This has the advantage to studies contrasting a (continuous) DBS condition to a no-DBS condition, in so far as DBS and no-DBS trials are unlikely to differ regarding the clinical state of patients or their overall alertness and task engagement. Stimulation was applied simultaneously to both hemispheres and ramped up and down to reduce paresthesia [25,80,81]. Ramp duration depended on the DBS intensity and ranged from 115 to 230 ms (see S1 Table). DBS intensity was titrated by slowly increasing the intensity of continuous DBS on each side and evaluating clinical effects on Parkinsonian symptoms as well as putative side effects, in particular paresthesia, by a trained clinician (DMH). When the threshold for clinical effects was reached the intensity was noted and, in case of side effects, slightly decreased. We evaluated this procedure by performing double-blind UPDRS-III scores (upper and lower limb bradykinesia, rigidity, and tremor scores) in continuous DBS ON versus OFF. This showed a consistent improvement in clinical scores on average from 27.4 to 20.2 ($t_{13}$ = 6.151, $P < 0.001$, d = 1.64, paired samples $t$ test) confirming that the chosen intensities were clinically effective. We then used this intensity for burst stimulation while patients performed the same experimental task as described above. None of the patients reported paresthesia during the experiment.

An overview of the analysis of STN LFPs and DBS is given in S7D Fig.

## Statistical analysis of STN LFPs

Based on previous studies [15–18], we had a clear a priori hypothesis about the spectral characteristics of STN activity relevant for force adaptations, namely the beta-band (approximately 13 to 30 Hz). To assess whether STN beta power in the current study was relevant for action evaluation and adaptation, we applied the following analyses:

## Feedback period

We aligned changes in STN beta power to the feedback cues (the Direction-cue was shown at a fixed interval of 1 s after the Value-cue) and applied a linear mixed-effects (LME) model (Matlab function *fitlme*) using single trial beta power as dependent variable and Value-feedback as well as Direction-feedback as predictors in the same model.

$$\mu_j = \beta_{0j} + \beta_1 * \text{Value} + \beta_2 * \text{Direction} \tag{2}$$

While the intersect was allowed to vary between each participant $j$ (random effect), the slopes were fixed effects. All single trial values of STN beta power were z-scored by subtracting the mean and dividing by the standard deviation for each patient. Trials with z-scores >3 were

excluded (<1% of trials). Value-feedback (shown at the first feedback-cue) ranged from 0 (worst) to 10 (best). Direction-feedback (second feedback-cue, i.e., too little versus too much force) was calculated as (10-value) multiplied by −1 (too little) or +1 (too much) resulting in a scalar ranging from −10 (force was >10% MVC lower than target force) to +10 (force was >10% MVC higher than target force), while 0 indicates no error.

We conducted these LME for 100 ms long moving windows of mean STN beta power, which were shifted by 10 ms from 0 (onset of Value-feedback) to 2 s (onset of Direction-feedback was at 1 s). The resulting t-values were then plotted over time, thresholded (corresponding to $p < 0.05$) and the resulting clusters, which consisted of all time points that exceeded the initial threshold, were compared against the probability of clusters occurring by chance by randomly shuffling the trial order of STN beta power using 1,000 permutations [82,83]. Of note, single trial beta power was shuffled across trials, while the order of time windows within each trial was preserved. Only clusters in the observed data that were larger than 95% of the distribution of clusters obtained in the permutation analysis were considered significant and marked as $P_{cluster} < 0.05$. These cluster-based permutation tests take into account that the statistical test (LME) was conducted multiple times (over multiple time points), thus correcting for multiple comparisons and are routinely applied in neurophysiology research. For more details, Maris and Oostenveld have provided a thorough overview of the topic [82]. Post hoc tests of the significant time window comprised (i) model comparison in which model evidence according to the BIC were compared between models only including one of each predictor and the combined model (lower values indicate better performance given model complexity where a BIC difference >2 is considered positive evidence, >6 strong evidence, and >10 very strong evidence [84]); (ii) excluding each individual participant at a time (while the others remained in the LME) and testing whether the effect remained significant; (iii) applying separate regression analyses for each individual participant (instead of an LME) and performing a 1 sample $t$ test on the Fisher r-to-z transformed regression slopes to test for significance; (iv) applying an alternative baseline correction (−500 to 0 ms relative to the Value-cue instead of the mean across the whole experiment); (v) using a "wide" bipolar montage as in the stimulation session; and (vi) excluding individual hemispheres were both contacts of the bipolar montage were not localized within an anatomical STN mask ($n$ = 4 hemispheres, see above).

## Movement period

We aligned STN beta power to peak force and conducted LMEs as described for "Feedback period" using predictors reflecting force adaptation in the same model. Predictors were the absolute change in force (i.e., how much the force was adapted from the previous to the current trial) and change in force (negative for a decrease in force and positive for an increase in force compared to the previous trial). For example, in a trial in which force was reduced by 5% MVC compared to the previous trial, the absolute change in force was 5 and the change in force was −5.

$$\mu_j = \beta_{0j} + \beta_1 * \text{absolute change in force} + \beta_2 * \text{ change in force} \qquad (3)$$

LMEs were conducted for 100-ms long time windows of mean STN beta power shifted by 10 ms from −1 s to peak force. Correction for multiple comparisons using cluster-based permutation tests and post hoc tests of significant windows were performed as described above.

In an exploratory post hoc analysis, we also analyzed whether task-related changes in the alpha and gamma band (see Results) differed depending on the dorso-ventral gradient of the recording electrodes. To this end, we additionally extracted alpha power and gamma power from the time windows in which they showed an increase in the main analysis (alpha power

from the Value-feedback until 500 ms after the Value-cue and gamma power from 300 ms before peak force until peak force, see Results) from the neighboring more ventral bipolar contact and then compared the extent of task-related modulation of that contact to the bipolar contact from the main analysis (based on a functional localizer, see above) using paired sample *t* tests. If in either hemisphere the contact from the functional localizer already was the most ventral bipolar contact, we used only the contact from the contralateral hemisphere instead of averaging across sides. In 2 patients, the best beta contact was the most ventral bipolar contact in both hemispheres, leaving 14 patients for this analysis. Finally, we also conducted LME analyses with alpha and gamma power from the time windows of interest as dependent variable (from the contacts based on the functional localizer) as described for beta power above.

### Effects of stimulation on force adaptation

Timing-specific effects of STN burst stimulation were analyzed using a moving-window approach [80]. Stimulation intensity at each sample was saved in the recording software and imported to Matlab along with the TTL pulse (signaling the response), downsampled to 1,000 Hz and binarized (0 for no stimulation, 1 for stimulation). Since intensities during ramping up and down of stimulation were below the clinically effective intensity, they were defined as no stimulation [80]. For each trial, we noted for 100-ms long time windows if stimulation was applied or not (at any point during that window). This time window was shifted by 10 ms over 2,000 ms (from 0 to +2,000 ms) for the feedback-aligned data and over 1,000 ms (from −1,000 ms to peak force) for the movement-aligned data. We also analyzed the percentage of trials in which stimulation was applied at any given time window, which confirmed that stimulation was applied between approximately 40 and 50% of trials at all time windows (S4B and S4C Fig).

Based on the findings from the LFP-regression analysis (see Results), we hypothesized that stimulation would modulate the absolute change in force. To test this, for each time window we computed the mean absolute change in force for all trials in which stimulation was applied and all trials in which stimulation was not applied. At the second level, i.e., in the across-subjects analysis, we then tested whether this measure was affected by stimulation by performing cluster-based permutation tests [82,83]. At each time window, we computed the effect of stimulation using a cluster-building threshold corresponding to $p < 0.05$ and the resulting clusters, which consisted of all time points that exceeded the initial threshold, were compared against the probability of clusters occurring by chance by randomly shuffling between stimulation labels (stimulation versus no stimulation) of each participant using 1,000 permutations. Only clusters in the observed data that were larger than 95% of the distribution of clusters obtained in the permutation analysis were considered significant and marked as $P_{cluster} < 0.05$. The time window in which DBS had significant behavioral effects after the Value-cue (see Results) was termed $DBS_{value}$, while the time window where DBS had significant behavioral effects aligned to the movement was termed $DBS_{move}$.

Since the effects of DBS during the feedback period fluctuated over time and were close to statistical significance also at later windows (at approximately 600 ms and approximately 1,000 ms, see Fig 4B), we directly compared $DBS_{value}$ to these windows (from time points crossing the cluster-building threshold) using *t* tests and correcting for the 2 comparisons using Bonferroni correction. A putative correlation between effects of $DBS_{value}$ and $DBS_{move}$ across participants was assessed using Pearson correlation. To test whether the effect of $DBS_{value}$ on absolute change in force on the next trial was dependent on subsequent stimulation in $DBS_{move}$, we divided trials into trials where stimulation was applied versus not applied during $DBS_{move}$ and then tested whether the effect of $DBS_{value}$ remained significant using *t* tests and

corrected for the 2 comparisons using Bonferroni correction. We also directly compared the 2 conditions (DBS$_{value}$ with versus without premovement stimulation) using a paired *t* test.

As further supplemental analyses, we repeated these analyses using change in force (rather than absolute change in force) and collected points (Value-feedback). We also compared whether the effects of DBS$_{value}$ and DBS$_{move}$ differed depending on whether the force on the previous trial had been too low or too high or whether the previous Value-feedback had been relatively high or low (after a median split at 5 points). It should be noted that any further sub-division of trials (which were already separated into stimulation versus no-stimulation trials) depending on Direction- and Value-feedback resulted in a relatively low number of trials (on average approximately 20 per condition and participant).

### Effects of burst stimulation on STN LFPs

While stimulation was applied, LFPs were continuously recorded through the 2 contacts neighboring the stimulation contact and a bipolar signal derived as previously described (i.e., wide bipolar recording). Despite common-mode rejection, the artifact was clearly visible (S5A Fig) and its spectral characteristics were not strictly confined to the stimulation frequency and its harmonics (S5B and S5C Fig). Hence, the following artifact removal procedure was applied. The data were imported to Matlab, high-pass filtered at 4 Hz and low-pass filtered at 100 Hz using a fourth-order Butterworth filter, demeaned and detrended (*ft_preprocessing*). After visual inspection of the LFPs from each patient, a common threshold was set at 10 μV. This was chosen, because the remaining (i.e., after filtering) stimulation artifact, but not physiological LFPs (in the interval of stimulation bursts), consistently crossed this threshold. At each sample, the signal was removed if it crossed the threshold (approximately 9% of the data) and replaced by linear interpolation of the neighboring non-noisy signals. Afterwards, the data were downsampled to 200 Hz and the subsequent time-frequency analysis was identical to the LFP recordings described above. An example of single subject beta power is shown in S5D and S5E Fig; subject-averaged spectra are shown in S5F and S5G Fig.

After preprocessing and artifact correction, we first assessed the overall effect of stimulation on beta power by aligning beta power to onset of stimulation (after ramping) and normalizing it to the mean beta power when no stimulation was applied (i.e., during the stimulation interval). As expected [26–28], this showed a clear stimulation-related reduction in beta power in the time period after stimulation onset (from 0 to 500 ms after the onset), see S6 Fig. To assess the spectral specificity of this effect, we also analyzed DBS-induced changes in the theta, alpha, and gamma frequency bands in this time window using one-sample *t* tests and applying Bonferroni correction.

Then, we analyzed the effects of timing-specific stimulation (during DBS$_{value}$ and DBS$_{move}$) on STN beta power. Since the LFP regression analysis showed a relationship between levels of STN beta power and behavioral changes (see Results) and since stimulation reduced beta power (S6 Fig), we asked whether stimulation at these specific time points also affected beta power at specific windows. To this end, we first extracted beta power from the feedback-period in which stimulation affected behavior (DBS$_{value}$) and where STN beta power usually (i.e., off stimulation) correlated with Value, i.e., from 180 to 700 ms after Value-feedback (gray rectangle in Fig 4D). We then compared trials in which stimulation had been applied in the critical time window (DBS$_{value}$) and compared this to trials in which stimulation also affected behavior but in a distinct window (DBS$_{move}$). The rationale for this was to match the 2 conditions as well as possible regarding recording technique and signal quality (both conditions were from the stimulation session with wide bipolar contacts). Thus, any differences could not be due to differences in electrode configuration or artifact correction method. We also plotted beta

power from the off-stimulation session as reference. However, it should be noted that in these trials, both recording technique (narrow bipolar) and signal quality (no stimulation artifacts) were different.

We conducted the analogous analysis for movement-aligned data comparing beta power of $DBS_{move}$ and $DBS_{value}$ in the time window from 680 to 300 ms before peak force. The statistical analyses were conducted using cluster-based permutation tests by shuffling between stimulation labels ($DBS_{move}$ versus $DBS_{value}$) as described above.

## DBS connectivity analysis

In a final analysis, we assessed spatial specificity of stimulation effects using a recently developed approach [29]. For each hemisphere, the volume of activated tissue around the active contact was computed based on an electric conduction model [30], the individual electrode position and individual stimulation parameters. Next, the stimulation volume was used as seed region for whole-brain connectivity analysis using a published connectome dataset in PD patients [31,32]. In other words, each voxel of each participant was assigned a value reflecting how strongly it was connected to the stimulation volume. Then, Spearman correlation analyses were performed using this connectivity value and the individual's behavioral effect of DBS on force adaptation (since there were 2 significant time windows we used the average of $DBS_{value}$ and $DBS_{move}$ for each patient). To statistically assess spatial specificity of DBS effects, we defined a cortical area that we a priori hypothesized to be related to stronger behavioral stimulation effects, i.e., primary motor cortex (defined by the precentral gyrus in the Harvard-Oxford whole-brain atlas, [85]), which has been shown to be involved in force production and adaptation [19,33,34], is structurally connected to STN (in particular to dorsal STN expressing STN beta activity [35]) and related to clinical improvement after STN-DBS in PD [36]. As a control site, we chose a region that also is structurally connected to the STN, but we hypothesized not to be involved in STN DBS effects on force adaptation, i.e., temporal cortex (defined by the middle temporal gyrus in the same atlas). Since we had a clear expectation about the direction of results (stronger connectivity to motor cortex should be related to stronger force adaptation), we used one-tailed $p$-values and Bonferroni corrected for the 2 correlations.

## Supporting information

**S1 Fig. Force production and adaptation. (A)** Single participant traces of force and yank (first derivate of force) aligned to peak force for HCs. **(B)** Same as A but for PD patients. **(C)** Mean actual force (black: PD patients; blue: HC) are plotted along with target force (dotted lines) for all trials of trajectory A (upper panel) and trajectory B (middle panel). Participants showed the strongest errors when the target force was very low (e.g., middle trials of trajectory A or last trials of trajectory B). This was most pronounced in PD patients as can be seen in the lower panel showing the absolute error over trials. MVC, maximum voluntary contraction. Shaded areas in A–C represent SEM. Underlying data can be found in Mat1 and Mat23 in S1 Data.
(TIFF)

**S2 Fig. Localization of recording electrodes. (A)** 3D view of reconstructed bilateral leads overlaid on an anatomical mask of the subthalamic nucleus (STN in orange color; the shading is due to the overlay with the axial brain slice). **(B–D)** Same as A in 2D space separately for sagittal (B), axial (C), and coronal (D) slices. Throughout the figure electrodes from which bipolar LFP signals were analyzed are marked in red (in A some are overlaid by others and therefore obscured). In 4 of 26 hemispheres, both contacts used for the bipolar montage were localized

outside the anatomical STN mask according to lead reconstruction (see also S1 Table and Methods). In B–D, STN is indicated by an orange mask, while blue, green, and purple masks indicate, respectively, external pallidum, internal pallidum, and red nucleus.
(TIFF)

**S3 Fig. Relationship between STN beta power and absolute change in force locked to the value cue.** As expected from the varying onsets of movement execution after the Go signal, the relationship between STN beta power and absolute change in force was not present when locking the data to onset of the Value-cue (shown for the 2 s window prior to onset of the Value-feedback). Underlying data can be found in Mat24 in S1 Data.
(TIFF)

**S4 Fig. Burst stimulation. (A)** DBS bursts from example trials illustrating when stimulation bursts (black rectangles) were applied in the first 10 trials of patient 1, aligned to peak force. In all patients, DBS was given in bursts of varying duration (mean 250 ms). Between stimulation bursts was an approximately 500-ms long pause (mean 150 ms interval + ramping up and down each lasting ca. 175 ms on average = 500 ms, see also S1 Table). **(B)** Stimulation occurred on approximately 50% of trials for any given 100 ms window throughout the feedback time period. **(C)** Same as A for peak force aligned data. **(D)** There were no significant effects of stimulation on change in force (positive values for increase in force, negative values for decrease in force) on the next trial when aligning data to the feedback cues. **(E)** Same as D for peak force aligned data. **(F)** There were no significant effects of stimulation on collected points (Value-feedback) on the next trial when aligning data to the feedback cues. **(G)** Same as F for peak force aligned data. Shaded areas in B and C represent SEM. Horizontal gray lines in D–G show the cluster-building threshold. DBS, deep brain stimulation. Underlying data can be found in Mat25-30 in S1 Data.
(TIFF)

**S5 Fig. Stimulation-induced artifact. (A)** During the stimulation session, local field potentials were recorded from bipolar contacts surrounding the stimulation electrode. Despite common mode rejection, the artifact was clearly visible in the unprocessed data. **(B)** Example of time-frequency spectrum without artifact correction (see Methods for details regarding artifact correction) for an example patient. The spectral properties of stimulation-related artifacts were not restricted to the stimulation frequency and its harmonics. **(C)** Same as B but after artifact correction. **(D)** When the artifacts were strongly expressed, as in this patient (from B), they obliterated the normal movement-related beta power modulation in the trial averaged data (peak force aligned). **(E)** After artifact correction, the normal (i.e., as observed in the off stimulation session) beta modulation can be seen in the trial averaged data (peak force aligned). **(F)** Group average of peak force aligned spectrum after artifact correction (compare to Fig 3A). **(G)** Group average of feedback aligned spectrum after artifact correction (compare to Fig 2A). LFP, local field potential. Underlying data can be found in Mat31-34 in S1 Data.
(TIFF)

**S6 Fig. Stimulation-induced changes in STN power. (A)** Group-averaged time-frequency spectrum (not normalized) aligned to onset of stimulation (after ramping). The dotted box indicates STN beta power after stimulation onset. For statistical tests of changes in different frequency bands, see S3 Table. **(B)** Beta power (extracted from dotted box in A) is aligned to onset of stimulation and normalized to the time period where no stimulation was applied. Stimulation led to an approximately 40% decrease in beta power, which returned to baseline after approximately 0.5 s (mean burst duration was 250 ms +/− 100 drawn from a uniform distribution). The thick black line indicates the group mean, and the shaded area around this line

represent SEM. The thin gray lines illustrate beta band changes from individual participants. DBS, deep brain stimulation. Underlying data can be found in Mat35-36 in S1 Data.
(TIFF)

**S7 Fig. DBS leads and analysis overview. (A)** Each DBS (deep brain stimulation) lead had 8 contacts (2 omnidirectional leads and 6 directional leads) on 4 vertical levels. **(B)** Nine bipolar pairs were computed between neighboring contacts on the vertical levels for the OFF stimulation session. **(C)** For the ON stimulation session, wide bipolar contacts were created and the intervening contact was used for applying burst stimulation. **(D)** Analysis overview: For the OFF stimulation session, 9 bipolar contacts per hemisphere were computed based on the 8 monopolar recording contacts. In each hemisphere, the bipolar pair with the most pronounced beta power changes during movement was chosen for further analyses, since this indicates proximity to or localization within dorsal STN and correlates with motor performance (see Methods) and averaged across hemispheres. Single trial values were extracted and regressed against behavioral variables of interest using LME models. During the ON stimulation condition, a wide bipolar contact (shown in C) was created in each hemisphere to mitigate stimulation artifacts using common-mode rejection. The wide bipolar pair showing the most pronounced beta power changes during movement (OFF stimulation) was chosen for recordings. Timing-specific DBS effects on behavior were assessed by analyzing force adaptation in time windows where stimulation was applied vs. not applied in a sliding-window approach. Corresponding DBS effects on STN activity were analyzed for the surrounding bipolar contacts.
(TIFF)

**S1 Table. Clinical details.** Age and disease duration are given in years. Clinical scores are given as total score of the MDS Unified Parkinson's disease rating scale (UPDRS) part III for levodopa ON/OFF and as items 3–8 and 14–18 (limb scores) for DBS ON/OFF. Medication is given in levodopa-equivalent daily dose (LEDD). All patients received levodopa, 12 patients received a dopamine-agonist, 10 patients a catechol-O-methyltransferase (COMT) inhibitor, 9 patients a monoamine-oxidase (MAO) inhibitor, and 4 patients amantadine. D and V indicate whether the respectively more dorsal or ventral contact was chosen as active contact in the left and right hemisphere, after which DBS intensity is given in mA and ramp time is given in seconds. The last column lists for each hemisphere whether at least one of the contacts of the bipolar montage used for the local field potential analysis overlaid with a structural mask of the STN (see Methods). n/a, not available.
(DOCX)

**S2 Table. Overview of behavioral group comparisons.** au, arbitrary units; AUC, area under the curve; CV, coefficient of variation; dof, degrees of freedom; dt, time derivative; ms, millisecond; MVC, maximum voluntary contraction; N, Newton; RMSE, root mean squared error; s, second. * These 2 effects were significantly different from each other when directly comparing them ($t_{29} = -2.661$, d = 0.956, $P = 0.013$). Significant effects are shown in bold.
(DOCX)

**S3 Table. DBS-induced changes in STN power for different frequencies.** dof, degrees of freedom. Significant effects are shown in bold.
(DOCX)

**S1 Text. Comparison of force adaptation in Parkinson's disease patients and healthy controls.**
(DOCX)

**S1 Data. Zip file showing data underlying reported results in separate mat-files for individual figure panels.** Mat1: Data for Fig 1C. Force (colums 1–1,001) and yank (columns 1,002–2,002) from −0.5 to +0.5 s aligned to peak force. Mat2: Data for Fig 1D. Average points (column 1), coefficient of variation (column 2) and by-trial change in force (column 3). Mat3: Data for Fig 2A. Struct containing powerspectrums of frequencies (2–100 Hz) * time (−3 to +3 s aligned to Value-feedback). Each cell corresponds to an individual participant. Mat4: Data for Fig 2B: Alpha power from −1 to +2 s aligned to Value-feedback. Mat5: Data for Fig 2C: Same as Mat4 for beta power. Mat6: Data for Fig 2D. Regression between beta power and Value-feedback (t-statistic from LME) from 0 to +2 s aligned to the Value-cue. Mat7: Data for Fig 2E. Same as Mat6 for Direction feedback. Mat8: Data for Fig 3A. Same as Mat3 for movement aligned data (−3 to +3 s aligned to peak force). Mat9: Data for Fig 3B. Same as Mat4 for movement aligned data (−1 to +2 s aligned to peak force) and gamma power. Mat10: Data for Fig 3C. Same as Mat9 for beta power. Mat11: Data for Fig 3D. Regression between beta power and absolute change in force (t-statistic from LME) from −1 to 0 s aligned to peak force. Mat12: Data for Fig 3E. Same as Mat11 for change in force. Mat13: Data for Fig 4B. DBS effects on absolute change in force from 0 to +2 s aligned to Value-feedback. Mat14: Data for Fig 4C. Absolute change in force without stimulation (first column) and with stimulation (second column) for DBS$_{value}$. Mat15: Data for Fig 4D. Beta power from the stimulation session from −1 to +2 s aligned to Value-feedback. Mat16: Data for Fig 4E. Beta power separately for DBS$_{move}$ (rows 1–14) and DBS$_{value}$ (rows 15–28) from 180 to 700 ms after Value-feedback. Mat17: Data for Fig 4F. Beta power during the significant time window from Fig 4E with stimulation during DBS$_{value}$ (first column) and DBS$_{move}$ (second column). Mat18: Data for Fig 5B. DBS effects on absolute change in force from −1 to 0 s aligned to peak force. Mat19: Data for Fig 5C. Absolute change in force without stimulation (first column) and with stimulation (second column) for DBS$_{move}$. Mat20: Data for Fig 5D. Beta power from the stimulation session from −1 to +2 s aligned to peak force. Mat21: Data for Fig 5E. Beta power separately for DBS$_{value}$ (rows 1–14) and DBS$_{move}$ (rows 15–28) from 680 to 300 ms before peak force. Mat22: Data for Fig 5F. Beta power during the significant time window from Fig 5E with stimulation during DBS$_{value}$ (first column) and DBS$_{move}$ (second column). Mat23: Data for S1C Fig. Actual force (columns 1–100), target force (columns 101–200), and absolute error (columns 201–300). NaNs refer to invalid trials. Mat24: Data for S3 Fig. Regression between beta power and absolute change in force (t-statistic from LME) from −2 to 0 s aligned to Value-cue. Mat25: Data for S4B Fig. %trials on stimulation from 0 to +2 s aligned to Value-feedback. Mat26: Data for S4C Fig. %trials on stimulation from −1 to +0 s aligned to peak force. Mat27: Data for S4D Fig. DBS effects on change in force from 0 to +2 s aligned to Value-feedback. Mat28: Data for S4E Fig. DBS effects on change in force from −1 to 0 s aligned to peak force. Mat29: Data for S4F Fig: DBS effects on collected points from 0 to +2 s aligned to Value-feedback. Mat30: Data for S4G Fig. DBS effects on collected points from −1 to 0 s aligned to peak force. Mat31: Data for S5D Fig. Frequency (2–100 Hz) * time (−3 to +3 s aligned to peak force) spectrum of one participant from the stimulation session without artifact correction. Mat32: Data for S5E Fig: Same as Mat31 with artifact correction. Mat33: Data for S5F Fig. Struct containing powerspectrum of frequencies (2–100 Hz) * time (−3 to +3 s aligned to peak force). Each cell corresponds to an individual participant. Mat34: Data for S5G Fig. Same as Mat33 for Value-feedback aligned data. Mat35: Data for S6A Fig. Struct containing powerspectrum of frequencies (2–45 Hz) * time (−0.24 to +0.76 s aligned to DBS onset). Each cell corresponds to an individual participant. Mat36: Data for S6B Fig. Beta power from −0.2 to +0.7 s aligned to DBS onset. All files containing data from healthy participants and patients (Mat1-2 and Mat23) have 31 rows (15 healthy controls followed by 16 patients). Note that data for S1A and S1B Fig is given in Mat1. (ZIP)

## Acknowledgments

For the purpose of Open Access, the author has applied a CC BY public copyright license to any Author Accepted Manuscript version arising from this submission.

## Author Contributions

**Conceptualization:** Damian M. Herz, Rafal Bogacz, Alek Pogosyan, Sergiu Groppa, Peter Brown.

**Formal analysis:** Damian M. Herz, Gabriel Gonzalez-Escamilla.

**Funding acquisition:** Damian M. Herz, Sergiu Groppa, Peter Brown.

**Investigation:** Damian M. Herz, Manuel Bange, Miriam Auer, Muthuraman Muthuraman, Martin Glaser, Sergiu Groppa.

**Supervision:** Muthuraman Muthuraman, Rafal Bogacz, Huiling Tan, Sergiu Groppa, Peter Brown.

**Writing – original draft:** Damian M. Herz.

**Writing – review & editing:** Manuel Bange, Gabriel Gonzalez-Escamilla, Miriam Auer, Muthuraman Muthuraman, Martin Glaser, Rafal Bogacz, Alek Pogosyan, Huiling Tan, Sergiu Groppa, Peter Brown.

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
