## [Editor Report · Decision Letter 0]

30 Nov 2022

Dear Dr Herz, 

Thank you for submitting your manuscript entitled "Neural underpinnings of action adaptation in the subthalamic nucleus." for consideration as a Research Article by PLOS Biology.

Your manuscript has now been evaluated by the PLOS Biology editorial staff, as well as by an academic editor with relevant expertise, and I am writing to let you know that we would like to send your submission out for external peer review.

Once your full submission is complete, your paper will undergo a series of checks in preparation for peer review. After your manuscript has passed the checks it will be sent out for review. To provide the metadata for your submission, please Login to Editorial Manager (https://www.editorialmanager.com/pbiology) within two working days, i.e. by Dec 02 2022 11:59PM.

Kind regards,

Kris

Kris Dickson, Ph.D., (she/her)

Neurosciences Senior Editor/Section Manager

PLOS Biology

kdickson@plos.org

---

## [Decision Letter · Decision Letter 1]

2 Mar 2023

Dear Dr Herz,

Thank you for your continued patience while your manuscript "Neural underpinnings of action adaptation in the subthalamic nucleus." was peer-reviewed at PLOS Biology. Please accept my sincere apologies for the long delays that you have experienced during the peer review process. I am now handling your manuscript since Kris Dickson has now moved on from PLOS Biology. Your manuscript has now been evaluated by the PLOS Biology editors, an Academic Editor with relevant expertise, and by two independent reviewers. 

In light of the reviews, which you will find at the end of this email, we would like to invite you to revise the work to thoroughly address the reviewers' reports.

As you will see, both reviewers are positive about the study but note that it is difficult to follow all of the steps in the analysis and suggest that schematic diagrams and additional paragraphs are included to provide clarity and context for the approach. In addition, Reviewer #2 also questions how general stimulation effects can be excluded and notes the lack of a sham stimulation condition. After discussions with the academic editor, we will not make sham control data with additional patients a requirement for publication, but the comments could be addressed using additional electrode contacts which are ineffective for STN stimulation or otherwise not used for the regular use case, which could then potentially serve as sham stimulation sites. Alternatively, a post-experiment debriefing could be used where patients are asked if and what they experienced during stimulation and then report this to have an impression of potential uncontrolled side effects. If additional data or information cannot be included in the revision, then we ask that you please directly address the topic of sham stimulation conditions in the Methods and Discussion sections. 

Given the extent of revision needed, we cannot make a decision about publication until we have seen the revised manuscript and your response to the reviewers' comments. Your revised manuscript is likely to be sent for further evaluation by all or a subset of the reviewers.

**IMPORTANT - SUBMITTING YOUR REVISION**

*Re-submission Checklist*

*Published Peer Review*

*PLOS Data Policy*

*Blot and Gel Data Policy*

Sincerely,

Richard

Richard Hodge, PhD

Associate Editor, PLOS Biology

rhodge@plos.org

REVIEWS:

Reviewer #1: Herz et al describe findings on action adaptation in 16 PD patients undergoing DBS. Decreases in subthalamic beta (13-30 Hz) activity reflected stronger action adaptation. Moreover, if beta activity was suppressed by STN DBS in a specific time window, this also led to stronger action adaptation suggesting that time-specific modulation of STN beta activity facilitates adaptive behavior.

This is a well-designed and thoroughly conducted study from a highly experienced group. The approach is innovative with the combination of beta activity recording and application of bursts of stimulation to explore the effects of time -specific beta band modulation on behavior. The study shows that dynamics of STN beta activity are related to action adaptation. 

Major points: 

Although the manuscript is well structured, I think it is quite difficult to follow all steps of analysis in the result section and it could be helpful to have a short paragraph that better explains the reasoning for the analysis steps including an overview figure. Moreover, the main result and its value for understanding action adaptation and adjusting DBS should be discussed more specifically.

Moreover, the authors state that "this provides further evidence for the usefulness of STN beta activity as a read-out or feedback signal for adaptive DBS approaches and demonstrates that precisely delivered electrical stimulation can modify movement adaptation in humans." In my mind it is far fetched that time specific adaptive DBS could be adjusted to modify action/movement adaptation. The authors do not know from their study if a general (adaptive) beta suppression has an influence on action adaptation or if beta suppression at different time points may even interfere with adaptation. 

The authors point out that they observed a decrease in beta activity after worse outcomes (other than expected with reduced dopamine release). This should be discussed also in the light of the dopaminergic treatment that was used in the study. I do not follow what they mean by "a more nuanced relationship between dopamine release and beta power". 

Additionally, authors have conducted their study ON medication and use this as an argument to have "normal" motor performance in their patients. However, interaction of levodopa treatment and the concept of reward-related dopamine release should be discussed. Moreover, it is known that beta activity is suppressed by levodopa treatment. Shouldn't the effects of STN DBS be more clear with patients OFF medication? 

I also do not follow the rational to argue that recordings immediately after surgery are of specific value. DBS stun effect has been shown to interfere with LFP recordings and that should be mentioned. 

BIC differences of the different models are quite similar. Could the authors comment on the relevance of difference of a few points. 

Reviewer #2: First of all, I would like to congratulate the authors on the approach they took and the amount of effort that must have gone into this study. Combining DBS recordings and stimulation within participants is in my view an excellent way to tackle the research questions of this project. I have some open questions, in particular regarding the specificity of the results, which I outline below. 

Major: 

1. General stimulation effects / lack of sham stimulation condition

How do the authors exclude general stimulation effects? They compare DBS vs. Off, but how do we know that it is the STN-specific DBS that is driving the observed effects and not a general stimulation effect that could be reached by stimulation anywhere in or even outside the brain? The authors do provide data on the temporal specificity of the stimulation (not all stimulations at all time points result in behavioral changes), but could they provide data on the spatial specificity as well? 

2. Electrode rereferencing and localization

-The re-referencing scheme is quite complex. I am worried that readers may not be familiar with the specific electrode model and its layout with multiple types of contacts. It would be helpful to have a sketch in the supplementary material, depicting an electrode with omni- and directional contacts and how these contacts are rereferenced against each other. It would be great to have sketches for recording contacts as well as for the stimulation contacts, since the referencing scheme was different.

- on pages 20/21, the authors describe that they picked the bipolar pair for further analyses based on movement related decrease in beta power and state that they "confirmed the validity of this functional localizer approach by conducting lead localization analysis", referring to Suppl. Fig. 5, which contains a depiction of the electrodes and an atlas. On p. 15 they say that the contacts "mainly localized to the 'motor' STN." From the figure alone, it is hard to tell whether and which the contacts that were picked via the functional localizer also fell into the STN target region. 

Could the authors provide this information in a table? Please also add information about inclusion/exclusion rules: e.g., did both or one contact that was merged into bipolar pair have to be in the target region, etc.? Adding another angle/plane to the depiction in Suppl. Fig. 5 might also help to better grasp the localization of the electrodes. 

Do the main results still hold if the authors only analyze recordings from bipolar pairs that show the largest decrease in beta power and at the same time at least one of the contacts fell into the 'motor' STN?

-What happens if the stimulation rereferencing scheme is used to build bipolar pairs for the recordings and main analyses are repeated? Do the main results still hold? 

3. Why do the authors use predefined frequency bands? E.g., alpha peak varies drastically with all kinds of factors, and will also be different in patients and healthy participants. Could the authors use a more data-driven approach to define their frequency bands? This might, for example, increase the signal for gamma power and reveal effects related behavior. 

4. Is it problematic that the onset of the Direction-Cue was predictable? Due to the constant time difference between both cues, some of the processes elicited by the onset of the cue (without having knowledge of the direction) might have shifted in time because they could be anticipated. 

5. Why did the authors choose the mean power across the whole recording as a baseline? Would the main results still hold if a baseline within the task was used, e.g., a period prior to value-cue onset?

6. Line 37 & Fig. 5D: Why was this exact time window from 680 to 300 ms pre peak-force chosen? Why not use the time window 440-300 ms pre peak-force as in Fig. 3D? Do the results still hold for the 440-300 ms time window?

Minor: 

-The authors could indicate the meaning of t=0 in Fig. 1C, in the figure legend. 

-Line 164/165: Was this corrected for multiple comparisons? If yes, please indicate in the text, if no, please do so (comparisons are conducted for multiple time bins, inflating false positives). At the very least, indicate that this is uncorrected.

-Please provide more information about the DBS electrodes: How many contacts per lead, how large is a contact, what is the distance between contacts. 

- Is the effect in beta power locked to peak-force (Fig. 3), also observable when locking the data to the value cue?

-Are the time-frequency plots in Fig. 2A and 3A informative? Since it is an average across participants, the depicted in-/decreases could be driven by single participants / outliers and potentially be driven by the type of baseline the authors used (mean power across the whole recording can vary drastically across patients). Could the authors add a depiction that takes variability across participants into account? 

-Looking at the time-frequency plot in Fig. 2A, there seems to be a prominent decrease before the onset of the direction-cue, in a somewhat lower frequency than the alpha increase in Fig. 2B (but see comment above). Do the authors have an idea what is going on there?

-Line 643: "dorsal STN [31, 32]" vs. line 674: "dorsal STN [13, 31]"

References correct?

---

## [Decision Letter · Decision Letter 2]

19 Apr 2023

Dear Dr Herz,

Thank you for your patience while we considered your revised manuscript "Neural underpinnings of action adaptation in the subthalamic nucleus." for publication as a Research Article at PLOS Biology. This revised version of your manuscript has been evaluated by the PLOS Biology editors, the Academic Editor and the original reviewers.

Based on the positive reviews, I am pleased to say that we are likely to accept this manuscript for publication, provided you satisfactorily address the following data and other policy-related requests that I have provided below (A-E):

(A) We would like to suggest the following modification to the title, to make it more compelling and accessible for our broad readership:

“Dynamic modulation of subthalamic nucleus activity facilitates adaptive behavior”

(B) You may be aware of the PLOS Data Policy, which requires that all data be made available without restriction: http://journals.plos.org/plosbiology/s/data-availability. For more information, please also see this editorial: http://dx.doi.org/10.1371/journal.pbio.1001797

Thank you for already providing the underlying data for the main figures in the file ‘SourceData’. However, we note that the underlying data for the Supplementary figures seems to be missing and we would be grateful if you could also provide the underlying data for the following figures:

Figure S1A-C, S3, S4B-G, S5B-G, S6A-B

(C) Please also ensure that each of the relevant figure legends in your manuscript include information on *WHERE THE UNDERLYING DATA CAN BE FOUND*, and ensure your supplemental data file/s has a legend.

(D) Please ensure that your Data Statement in the submission system accurately describes where your data can be found and is in final format, as it will be published as written there. Specifically, we ask that you please update the statement to reference the underlying data that can be found in ‘SourceData’.

(E) Thank you for noting that the minimum example dataset will be deposited at https://data.mrc.ox.ac.uk. We would be grateful if you could provide more details about how the dataset can be accessed and provide the DOI of the deposition.

We expect to receive your revised manuscript within two weeks. 

*Published Peer Review History*

*Press*

Kind regards,

Richard

Richard Hodge, PhD

Associate Editor, PLOS Biology

rhodge@plos.org

Reviewer remarks:

Reviewer #1: The authors addressed all questions. Congratulations to the study.

Reviewer #2: The authors did a great job answering all my questions. I have no further concerns.

---

## [Editor Report · Decision Letter 3]

26 Apr 2023

Dear Damian,

Thank you for the submission of your revised Research Article "Dynamic modulation of subthalamic nucleus activity facilitates adaptive behaviour" for publication in PLOS Biology. On behalf of my colleagues and the Academic Editor, Alexander Gail, I am pleased to say that we can accept your manuscript for publication, provided you address any remaining formatting and reporting issues. These will be detailed in an email you should receive within 2-3 business days from our colleagues in the journal operations team; no action is required from you until then. Please note that we will not be able to formally accept your manuscript and schedule it for publication until you have completed any requested changes.

PRESS

Kind regards,

Richard

Richard Hodge, PhD

Associate Editor, PLOS Biology

rhodge@plos.org

PLOS
